# Polyhedron Attention Module: Learning Adaptive-order Interactions

**Tan Zhu**
University of Connecticut
tan.zhu@uconn.edu

**Fei Dou**
University of Georgia
fei.dou@uga.edu

**Xinyu Wang**
University of Connecticut
xinyu.wang@uconn.edu

**Jin Lu**
University of Georgia
jin.lu@uga.edu

**Jinbo Bi**
University of Connecticut
jinbo.bi@uconn.edu

## Abstract

Learning feature interactions can be the key for multivariate predictive modeling. ReLU-activated neural networks create piecewise linear prediction models. Other nonlinear activation functions lead to models with only high-order feature interactions, thus lacking of interpretability. Recent methods incorporate candidate polynomial terms of fixed orders into deep learning, which is subject to the issue of combinatorial explosion, or learn the orders that are difficult to adapt to different regions of the feature space. We propose a Polyhedron Attention Module (PAM) to create piecewise polynomial models where the input space is split into polyhedrons which define the different pieces and on each piece the hyperplanes that define the polyhedron boundary multiply to form the interactive terms, resulting in interactions of adaptive order to each piece. PAM is interpretable to identify important interactions in predicting a target. Theoretic analysis shows that PAM has stronger expression capability than ReLU-activated networks. Extensive experimental results demonstrate the superior classification performance of PAM on massive datasets of the click-through rate prediction and PAM can learn meaningful interaction effects in a medical problem.

## 1 Introduction

Learning feature interactions provides insights into how predictors (features in $\mathbf{x}$) interact to vary a dependent variable $y$, and could significantly improve prediction performance in a wide range of research areas, such as recommendation systems [1, 2, 3, 4], genetic analysis [5, 6] and neuroscience [7, 8], and help explain the decision-making of a complex model.

Interactions among features can significantly improve the model's expression capability. For a simple example, by incorporating the two-way interaction effect between gender (0/1: female and male) and age into the linear regression model $height \sim w_1 \cdot gender + w_2 \cdot age + w_3 \cdot gender \times age + w_0$ (where $w_0, w_1, w_2, w_3$ are trainable parameters), the effects of female's age on the heights will be different from that of male ($w_2$ v.s. $w_2 + w_3$). A predictive model $f(\mathbf{x})$ has a $k$-way interaction effect if it satisfies [9]:

**Definition 1 ($k$-way interaction effect)** *Let $\mathbf{x}$ be the input of a function $f(\mathbf{x}) : \mathbb{R}^p \to \mathbb{R}$, and $x_i$ be the $i^{th}$ feature of $\mathbf{x}$. Let $\mathcal{I} \subseteq \{1, 2, ..., p\}$ be a set of feature indices and $k$ ($k > 2$) be the cardinality of $\mathcal{I}$. If*

$$\mathbb{E}_{\mathbf{x}}\left[\frac{\partial^{|\mathcal{I}|} f(\mathbf{x})}{\partial x_{i_1} \partial x_{i_2} ... \partial x_{i_k}}\right]^2 > 0, \tag{1}$$

37th Conference on Neural Information Processing Systems (NeurIPS 2023).

*$f(\mathbf{x})$ exhibits a $k$-way interaction effect among features indexed by $i_1, i_2, ..., i_k \in \mathcal{I}$.*

A $k$-th order polynomial $f(\mathbf{x})$ can define $k$-order interactions which may not be $k$-way. A $k$-way interaction must occur among $k$ different features. For instance, a 2-way interaction between $x_i$ and $x_j$ may occur as $x_i x_j$ or $x_i^2 x_j^3$, so can be generally written as $\sum_{m_i, m_j} w_{m_i, m_j} x_i^{m_i} x_j^{m_j}$ for some nonzero $m_i$ and $m_j$ if coefficients $w$'s are nonzero. However, high dimensional features can generate enormous candidate interactions due to the combinatorial explosion, causing the curse of dimensionality. For example, $k$ binary features can have $2^k$ possible $k$-way interactions. This challenge in learning feature interactions attracts tremendous research interest in various models ranging from logistic regression to deep neural networks (DNN).

Early methods learn the effects of feature interaction by multiplying features together and employing the multiplications as input to create an additive model. Two lines of methods have been used to reduce the quantity of candidate interactions. Penalization methods (e.g., Hierarchical LASSO [10, 11] and Group-LASSO [12]) assume a hierarchy of interactions, so high-way interactions may be eliminated by removing low-way interactive terms. Factorization Machines (FM) require fewer parameters to fit all predefined interaction terms, which implicitly selects among candidate interactions [13, 14, 15]. Only 2- and/or 3-way interactions were considered by these methods. To learn interactions among more features (higher-way), models such as FM-supported NN [16], product-based NN [17], AutoInt+ [18], DESTINE [19] and AFN [20] predict the target with sigmoidal-activated (such as the softmax or sigmoid function) DNNs. However, sigmoidal-activated DNN models only capture extremely high-order interactions among all features according to Definition 1, because sigmoidal activation functions have continuous (non-zero) derivatives up to infinite order with respect to all input features.

To capture interaction effects among a subset of features, an increasing number of works adopt ReLU-activated plain DNN which fits a piecewise linear function [21], as the backbone to develop interaction learning models. Models such as MaskNet [22] first form interaction effects and then use them as the input of a ReLU-activated DNN, also known as the stacked structure. Recently, more state-of-the-art performance has been achieved with models such as DeepIM [23], DeepFM [24], DCN [25], xDeepFM [26], AOANet [4], DCN-V2 [25], EDCN [27] and FinalMLP [28] in a parallel structure, which combines outputs of the proposed feature interaction learning models with those of a ReLU-activated plain DNN by operators such as summation, concatenation or Hadamard product.

By fusing ReLU-activated DNNs with models that explicitly incorporate interaction effects, piecewise polynomial functions can be formed. Rather than predicting the target with a highly non-linear function infinitely differentiable on the input space, these methods divide the input space into pieces (e.g., polyhedrons), map instances to different pieces, and predict the target with a piece-specific polynomial function. For all existing models, the piece-specific polynomial functions fit the same format of interactions (fixed $\prod x_{i_1}^{m_1} x_{i_2}^{m_2} \cdot x_{i_k}^{m_k}$ terms for a specific $k$) across all pieces once they are identified. However, data instances belonging to different pieces may endorse interactions in different ways. To address this problem, we propose a new type of attention module called Polyhedron Attention Module (PAM). The main contribution of this paper are summarized as follows:

- PAM splits the input space into polyhedrons and predicts the target using a piecewise polynomial function which contains an attention term for each piece. The attention captures interaction effects of adaptive orders for different polyhedrons according to their local structure.

- We propose a model interpretation approach for PAM to identify important interaction effects for any given data instance.

- We prove a universal approximation theorem for PAM, and show that DNNs incorporating PAMs need fewer parameters than ReLU-activated plain DNNs to maintain the same level of approximation error in fitting functions in the Sobolev space.

- Empirical studies are designed to validate PAM on benchmark datasets that are previously recognized to require feature interaction learning, such as the massive Criteo (33 million samples, 2.1 million features) and Avazu (28.3 million samples, 1.5 million features) click-through-rate datasets. Using a medical database with feature meanings, we explore the interpretation of PAM.

## 2   Motivation - Reinterpreting ReLU-based DNN

We argue that a plain fully-connected DNN with ReLU activation function may be interpreted as an attention mechanism. Such a neural network contains $L$ consecutive blocks each constituting

a fully-connected layer and a ReLU-activation layer. Let $\mathbf{x}^\ell$ be the input of the $\ell$-th block which outputs a nested function $\text{ReLU}(W^\ell \mathbf{x}^\ell + b^\ell)$. The $\text{ReLU}(z)$ returns $z$ when activated (if $z \geq 0$) or $0$ otherwise. The consecutive $L$ blocks together create piecewise linear functions (each output in layer $L$ defines such a function). For a given raw input $\mathbf{x}$, let $I^\ell$ contains the indices of the activated ReLU functions in layer $\ell$ when $\mathbf{x}$ passes through the layer, and $W_{I^\ell}$ and $b_{I^\ell}$ contain the original weights, respectively, in $W^\ell$ and $b^\ell$ for those rows indexed in $I^\ell$ and 0 vectors for those in the complement set of $I^\ell$ (denoted by $I^{\ell-}$). Then, at a layer $\ell$, $W^{(\ell)}\mathbf{x} + b^{(\ell)} \geq 0$ where $W^{(\ell)} = \prod_{j=1}^{\ell} W_{I^j}$ and $b^{(\ell)} = \sum_{j=1}^{\ell} b_{I^j} \prod_{k=j+1}^{\ell} W_{I^k}$. More precisely, it means that $\mathbf{x}$ stays in a polyhedron that is defined by the intersection of all the half-spaces $\{\mathbf{x} : W^{(\ell)}\mathbf{x} + b^{(\ell)} \geq 0\}$ and $\{\mathbf{x} : W^{(\ell-)}\mathbf{x} + b^{(\ell-)} < 0\}$ for all $\ell = 1, \cdots, L-1$ where $W^{(\ell-)} = W_{I^{\ell-}} \prod_{j=1}^{\ell-1} W_{I^j}$ and $b^{(\ell-)} = b_{I^{\ell-}} + \sum_{j=1}^{\ell-1} b_{I^j} W_{I^{\ell-}} \prod_{k=j+1}^{\ell-1} W_{I^k}$ specify those affine functions not activated in layer $\ell$. Here, $W_{I^{\ell-}}$ and $b_{I^{\ell-}}$ contain the original weights in $W^\ell$ and $b^\ell$ for those rows indexed in $I^{\ell-}$ and 0 vectors for those in $I^\ell$. For all $\mathbf{x}$ in this polyhedron, the $L$-th layer outputs affine functions $W^{(L)}\mathbf{x} + b^{(L)}$.

Precisely, the input space is divided into non-overlapping polyhedrons and each polyhedron $\Delta$ is defined by a combination of activated ReLU functions across the layers (a formal proof is given in Appendix A). To ease the notation, we simply use $\Delta$ to denote the set of indices of the activated ReLU across all layers that identify the polyhedron and $\Delta^-$ denotes the index set of the inactivated ReLU in the layers. We use an indicator function $\mathbb{1}(\mathbf{x} \in \Delta)$ to define the polyhedron which returns 1 for all vectors $\mathbf{x}$ that satisfy $W_\Delta^{(\ell)}\mathbf{x} + b_\Delta^{(\ell)} \geq 0$ and $W_\Delta^{(\ell-)}\mathbf{x} + b_\Delta^{(\ell-)} < 0$ for $\ell = 1, \cdots, L-1$, and 0 otherwise. The $i$-th activated ReLU in the $L$-th layer corresponds to a piecewise linear function that computes $W_{\Delta,i}^{(L)}\mathbf{x} + b_{\Delta,i}^{(L)}$ on each piece $\Delta$. Note that $W_{\Delta,i}^{(L)}$ and $b_{\Delta,i}^{(L)}$ vary for different polyhedrons due to the differences in the ReLU activation in the early layers (illustrated in

Figure 1: An example of 2-layer ReLU-activated plain DNN with 1 output (the green shading shows the function value). Black lines are fitted in layer 1 (their intersection defines the polyhedrons) and the red line is fitted in layer 2 and varies on different polyhedrons.

Fig. 1). Denote the hyperplane $\{\mathbf{x} : W_{\Delta,i}^{(L)}\mathbf{x} + b_{\Delta,i}^{(L)} = 0\}$ by $H_{\Delta,i,L}$, which further splits $\Delta$ into two polyhedrons $\Delta_1 = \mathbb{1}(\mathbf{x} \in \Delta, W_{\Delta,i}^{(L)}\mathbf{x} + b_{\Delta,i}^{(L)} \geq 0)$ and $\Delta_2 = \mathbb{1}(\mathbf{x} \in \Delta, W_{\Delta,i}^{(L)}\mathbf{x} + b_{\Delta,i}^{(L)} < 0)$. Thus, this piecewise linear function can be written as:

$$f(\mathbf{x}) = \sum_\Delta \mathbb{1}(\mathbf{x} \in \Delta) \cdot \text{ReLU}(W_{\Delta,i}^{(L)}\mathbf{x} + b_{\Delta,i}^{(L)}) = \sum_\Delta \mathbb{1}(\mathbf{x} \in \Delta) \cdot \text{ReLU}\left(\frac{W_{\Delta,i}^{(L)}\mathbf{x} + b_{\Delta,i}^{(L)}}{||W_{\Delta,i}^{(L)}||} ||W_{\Delta,i}^{(L)}||\right)$$

$$= \sum_\Delta \underbrace{\mathbb{1}(\mathbf{x} \in \Delta_1)\text{dist}(\mathbf{x}, H_{\Delta,i,L})}_{\text{attention}} \underbrace{||W_{\Delta,i}^{(L)}||}_{\text{value on }\Delta_1} + \underbrace{\mathbb{1}(\mathbf{x} \in \Delta_2)\text{dist}(\mathbf{x}, H_{\Delta,i,L})}_{\text{attention}} \underbrace{0}_{\text{value on }\Delta_2}$$

$$(2)$$

where $\text{dist}(\mathbf{x}, H_{\Delta,i,L})$ means the distance from $\mathbf{x}$ to the hyperplane $H_{\Delta,i,L}$. The values on the two pieces $\Delta_1$ and $\Delta_2$ are constant $||W_{\Delta,i}^{(L)}||$ where $||\cdot||$ is the $\ell_2$ vector norm or 0. The attention is a function of $\mathbf{x}$ and depends on how far $\mathbf{x}$ is from one of the hyperplanes that define the polyhedron boundary (see Fig. 1 for illustration).

We observe that a polyhedron $\Delta$ is defined using a sequence of hyperplanes corresponding to the affine functions in different layers, but the attention of ReLU-activated DNNs is calculated based only on the hyperplane in the last layer for the polyhedron (piece). Although not all of the hyperplanes in early layers make the active boundary of a polyhedron (i.e., the polyhedron can locate in the interior of a half-space), using only one hyperplane in the attention is restrictive. An attention mechanism that allows multiple active boundary hyperplanes of a polyhedron to aid attention calculation may increase the model's expression power. Let $\mathcal{H}_\Delta$ contain all of the active boundary hyperplanes $H$ of $\Delta$. For convenient notation, and with mild relaxation, we rescale the $W_H$ to $-W_H$ for those inactivated affine functions in the DNN so the half-spaces can all be written in the form of $W_H\mathbf{x} + b_H \geq 0$. Then, $\mathbb{1}(\mathbf{x} \in \Delta) = \prod_{H \in \mathcal{H}_\Delta} \mathbb{1}(W_H\mathbf{x} + b_H \geq 0)$. To learn feature interaction effects, we multiply the distances from $\mathbf{x}$ to each hyperplane in $\mathcal{H}_\Delta$. Given a hyperplane is linear in terms of $\mathbf{x}$, multiplying the distances from $\mathbf{x}$ to two $(m)$ hyperplanes creates quadratic ($m$-th order) terms. Thus, the number of active boundary hyperplanes of $\Delta$ offers the upper bound on the order of the multiplicative terms

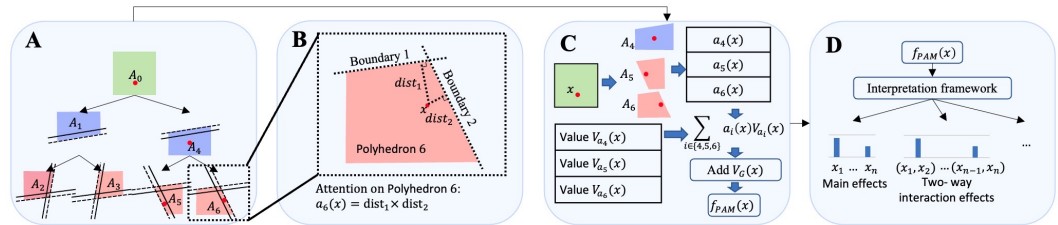

Figure 3: Overview description of the proposed method: (A) Split the input space into overlapping polyhedrons with the oblique tree. (B) Calculate attention on each polyhedron based on distances to the polyhedron's boundaries. (C) Calculate the model's output based on the attention and value vector. (D) Extract main and interaction effects from the model.

if each hyperplane is used only once. To allow the order to be adaptive to each $\Delta$ within the upper limit, we further enclose a bounding constraint on the distance (other strategies may exist, which we leave for future investigation). Thus, the new attention formula can be written as follows:

$$
\begin{aligned}
a_\Delta(\mathbf{x}) &= \mathbb{1}(\mathbf{x} \in \Delta) \prod_{H \in \mathcal{H}_\Delta} \min(\text{dist}(\mathbf{x}, H), \tfrac{2U_H}{||W_H||}) \\
&= \prod_{H \in \mathcal{H}_\Delta} \mathbb{1}(W_H \mathbf{x} + b_H \geq 0) \min(\text{dist}(\mathbf{x}, H), \tfrac{2U_H}{||W_H||}) \\
&= \prod_{H \in \mathcal{H}_\Delta} \max\left(\min\left(\tfrac{W_H \mathbf{x} + b_H}{||W_H||}, \tfrac{2U_H}{||W_H||}\right), 0\right),
\end{aligned}
\tag{3}
$$

where $U_H$ determines an upper bound on the distance from $\mathbf{x}$ to $H$ and is a trainable parameter. Fig. 2 demonstrates how adding the upper bounds $U_H$ allows the attention module to learn interaction effects of adaptive orders. For example, the instances in the area marked by 0 in the figure (left) are far from each boundary hyperplane $H$ beyond their respective upper bounds, so the min operator returns a constant $\frac{2U_H}{||W_H||}$. These $\mathbf{x}$ instances thus receive a constant attention that is the multiplication of these upper bounds each for an $H \in \mathcal{H}_\Delta$, which gives a 0-order interaction. When $\mathbf{x}$ is close to two of the hyperplanes (in those areas labeled by 2), two distance terms are used in Eq. 3, defining two-way interactions.

Early methods pre-specify a set of polynomial terms and use them as input features to a DNN [23] which limits the search space. In another line of methods [20], each feature is associated with an order parameter $m_i$ and products in the form of $\prod x_i^{m_i}$ are used as input (rather than raw features $x_i$) to a ReLU-activated DNN. The DNN is required to also learn the proper values of $m_i$. It is an excellent approach to identify useful interactions, but once these $m_i$'s are determined, the orders of feature interaction are fixed and cannot adapt to different pieces because the piecewise linear model (linear in terms of the input products) learned by the DNN does not change interaction orders. Unlike

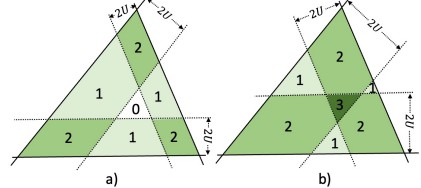

Figure 2: For a triangular polyhedron $\Delta$ in this example, the attention $a_\Delta$ can capture constant, additive, 2-way interactive, and 3-way interactive effects in the areas, respectively, marked by 0, 1, 2, and 3, by adjusting the upper bound parameter $U$ which is smaller in the left figure than in the right.

these methods, our attention module automatically identifies the interactions (polynomial terms) of appropriate orders according to the topology of a piece (particularly, using the number of active boundary hyperplanes of the polyhedron). Using introduced parameters $U_H$, we can learn the appropriate order of the interactive terms within a polyhedron by adjusting $U_H$ (larger values of $U_H$ lead to higher-orders of feature interaction). Our theoretical analysis (Section 5) shows that a DNN incorporating our attention module can approximate any function represented by ReLU-activated DNN at any arbitrarily accurate level with fewer model parameters.

## 3   The Proposed Polyhedron Attention Module (PAM)

This section elaborates on the PAM with the attention score defined in Eq. 3. With input $\mathbf{x} \in \mathbb{R}^p$, a DNN using PAM defines a function $f_{PAM}$:

$$
f_{PAM}(\mathbf{x}) = V(\mathbf{x}; \theta_G) + \sum_\Delta a_\Delta(\mathbf{x}) V(\mathbf{x}; \theta_\Delta),
\tag{4}
$$

where $V(\mathbf{x}; \theta_G)$ is a global value function with trainable parameter $\theta_G$, and $V(\mathbf{x}; \theta_\Delta)$ is the local value function on a piece $\Delta$ with trainable parameters $\theta_\Delta$. We set both the global and local value

functions to be affine functions, so $\theta$ contains the normal vector $W$ and offset $b$. Sec 3.2 explains why affine functions are our choice. Instead of forming polyhedrons by intersecting hyperplanes as done in ReLU-activated DNNs, we use a tree search to partition the input space into overlapping polyhedrons. Note that the potential set of partitions created by tree search is a superset of that created by hyperplane intersections (see Appendix B). We introduce how we generate polyhedrons and then discuss the attention and value functions.

## 3.1 Generating polyhedrons via oblique tree

Let $\mathcal{S}_\Delta$ contain all the polyhedrons needed to form a partition of the input space. We adopt the oblique tree to generate $\mathcal{S}_\Delta$. An oblique tree is a binary tree where each node splits the space by a hyperplane rather than by thresholding a single feature. The tree starts with the root of the full input space $\mathcal{S}$, and by recursively splitting $\mathcal{S}$, the tree grows deeper. For a $D$-depth ($D \geq 3$) binary tree, there are $2^{D-1} - 1$ internal nodes and $2^{D-1}$ leaf nodes. As shown in Fig. 3A, each internal and leaf node maintains a sub-space representing a polyhedron $\Delta$ in $\mathcal{S}$, and each layer of the tree corresponds to a partition of the input space into polyhedrons. Denote the polyhedron defined in node $n$ by $\Delta_n$, and the left and right child nodes of $n$ by $n_L$ and $n_R$. Unlike classic oblique trees that partition $\mathcal{S}$ into non-overlapping sub-spaces, we perform soft partition to split each $\Delta_n$ into $\Delta_{n_L}$ and $\Delta_{n_R}$ with an overlapping buffer. Let the splitting hyperplane be $\{\mathbf{x} \in \mathbb{R}^p : W_n\mathbf{x} + b_n = 0\}$. Then the two sub-spaces $\Delta_{n_L}$ and $\Delta_{n_R}$ are defined as follows:

$$\begin{aligned}
\Delta_{n_L} &= \{\mathbf{x} \in \Delta_n | \ W_n\mathbf{x} + b_n \geq -U_n\}, \\
\Delta_{n_R} &= \{\mathbf{x} \in \Delta_n | \ -W_n\mathbf{x} - b_n \geq -U_n\},
\end{aligned} \tag{5}$$

where $U_n$ indicates the width of the overlapping buffer. Eq. 5 shows that those instances satisfying $|W_n\mathbf{x} + b_n| < U_n$ belong to the buffer in both $\Delta_{n_L}$ and $\Delta_{n_R}$. This buffer creates a symmetric band around the splitting hyperplane.

Let $\mathcal{P}_n$ be the node set containing all the ancestor nodes above $n$. We can group the nodes in $\mathcal{P}_n$ into two subsets $\mathcal{P}_n^l$ or $\mathcal{P}_n^r$ where $n$ appears in the left or right subtree of those ancestors. Let $i$ index the nodes in $\mathcal{P}_n^l$ and $j$ index the nodes in $\mathcal{P}_n^r$. Then for any node $n$ except the root node, $\Delta_n$ can be expressed as the intersection of the half-spaces:

$$\Delta_n = \{\mathbf{x} \in \mathbb{R}^p : \ W_i\mathbf{x} + b_i \geq -U_i, \forall i \in \mathcal{P}_n^l, \text{ and } -W_j\mathbf{x} - b_j \geq -U_j, \forall j \in \mathcal{P}_{n^r}\}. \tag{6}$$

Based on the polyhedron defined by Eq. 6, the attention $a_{\Delta_n}(\mathbf{x})$ (which we refer to as $a_n(\mathbf{x})$ to simplify notation) can be rewritten as

$$a_n(\mathbf{x}) = \prod_{i \in \mathcal{P}_n^l} \max(\min(\tfrac{W_i\mathbf{x}+b_i+U_i}{||W_i||}, \tfrac{2U_i}{||W_i||}), 0) \prod_{i \in \mathcal{P}_n^r} \max(\min(\tfrac{-W_i\mathbf{x}-b_i+U_i}{||W_i||}, \tfrac{2U_i}{||W_i||}), 0), \tag{7}$$

where we use the buffer width $U_i$ to bound the corresponding distance term from above by $\frac{2U_i}{||W_i||}$.

Let $\mathcal{N}_d$ consist of all the nodes at the $d^{th}$ layer of the tree, $d \in \{1, 2, ..., D\}$. The nodes in $\mathcal{N}_d$ altogether specify a soft partition of the input space that maps an instance $\mathbf{x}$ to one sub-space or an intersection of overlapping sub-spaces in $|\mathcal{N}_d|$. Rather than merely utilizing the instance partitioning map defined by the last layer of the tree ($2^{D-1}$ polyhedrons), we allow PAM to leverage polyhedrons generated at all layers in $f_{PAM}$ with $\mathcal{S}_\Delta = \cup_{d=2}^D \{\Delta_n | n \in \mathcal{N}_d\}$ which gives $2^D - 2$ polyhedrons.

## 3.2 Learning the attention and value functions

Each internal node $n$ in the oblique tree needs to learn two affine functions: a splitting function used to form a hyperplane to split the sub-space into child nodes, and a value function. Because nested affine functions still produce affine functions, the set of affine functions is closed under linear transformations. Thus, we can use the value function to absorb the denominator $||W_i||$ from the attention Eq.7. In other words, for any learned $\theta_n$, we can find a $\theta_n'$ (by rescaling $\theta_n$ with those $\frac{1}{||W_i||}$) such that $a_n(\mathbf{x})V_n(\mathbf{x}, \theta_n) = a_n'(\mathbf{x})V_n(\mathbf{x}, \theta_n')$ where $a_n'(\mathbf{x}) = \prod_{i \in \mathcal{P}_n^l} \max(\min(W_i\mathbf{x} + b_i + U_i, 2U_i), 0) \prod_{i \in \mathcal{P}_n^r} \max(\min(-W_i\mathbf{x} - b_i + U_i, 2U_i), 0)$ and we use the subscript $n$ to denote the polyhedron $\Delta$ represented in node $n$. We thus directly set off to learn $\theta_n'$ and use $a_n'$ as $a_n$.

More importantly, with affine value functions, we can derive another property. For any internal node $n$, the attention of its two child nodes contains $a_n$ as a factor, so $a_{n_L}(\mathbf{x}) = a_n(\mathbf{x}) \max(\min(W_n\mathbf{x} +$

$b_n + U_n, 2U_n), 0)$ and $a_{n_R}(\mathbf{x}) = a_n(\mathbf{x}) \max(\min(-W_n\mathbf{x} - b_n + U_n, 2U_n), 0)$. It gives rise an observation that no matter where $\mathbf{x}$ is located (inside the buffer or outside), $a_{n_L}(\mathbf{x}) + a_{n_R}(\mathbf{x}) = 2U_n a_n(\mathbf{x})$. Thus, we can substitute $a_{n_L}(\mathbf{x})$ in the model $f_{PAM}$ by $2U_n a_n(\mathbf{x}) - a_{n_R}(\mathbf{x})$. Then, $a_n(\mathbf{x})V(\mathbf{x}, \theta_n) + a_{n_L}(\mathbf{x})V(\mathbf{x}, \theta_{n_L}) + a_{n_R}(\mathbf{x})V(\mathbf{x}, \theta_{n_R}) = a_n(\mathbf{x})\left(V(\mathbf{x}, \theta_n) + 2U_n V(\mathbf{x}, \theta_{n_L})\right) + a_{n_R}(\mathbf{x})\left(V(\mathbf{x}, \theta_{n_R}) - V(\mathbf{x}, \theta_{n_L})\right)$, which can be written as $a_n(\mathbf{x})V(\mathbf{x}, \bar{\theta}) + a_{n_R}(\mathbf{x})V(\mathbf{x}, \tilde{\theta})$ for some parameters $\bar{\theta}$ and $\tilde{\theta}$ due to the closure of affine functions under linear transformations. Hence, once again, we directly learn $\bar{\theta}$ and $\tilde{\theta}$ in our model training process. Note that we can recursively apply the above subsitution for all internal nodes $n$, so we can reduce the polyhedrons in $\mathcal{S}_\Delta$ by half. The following theorem characterize the above discussion (proof is in Appendix C.)

**Theorem 1** *If all value functions $V$ belong to a function set that is closed under linear transformations, then the function learned by PAM $f_{PAM}$ can be equivalently written as*

$$f_{PAM}(\mathbf{x}) = V(\mathbf{x}, \theta_G) + \sum_{n \in \mathcal{S}_\Delta^-} a_n(\mathbf{x})V(\mathbf{x}, \theta_n) \tag{8}$$

*where the polyhedron set $\mathcal{S}_\Delta^-$ contains half of the polyhedrons (e.g., the right child nodes or the left child nodes) in $\mathcal{S}_\Delta$ and*

$$a_n(\mathbf{x}) = \prod_{i \in \mathcal{P}_n^l} \max(\min(W_i\mathbf{x} + b_i + U_i, 2U_i), 0) \prod_{i \in \mathcal{P}_n^r} \max(\min(-W_i\mathbf{x} - b_i + U_i, 2U_i), 0). \tag{9}$$

**Remark 1** *Although we include polyhedrons identified by the internal nodes in our calculation, $f_{PAM}$ only needs to learn $\frac{1}{2}(2^D - 2) = 2^{D-1} - 1$ value functions, which is actually in the same scale as that of only using leaf nodes in $f_{PAM}$.*

**Optimization of PAM.** The output of $f_{PAM}(\mathbf{x})$ can be treated as a prediction of $\mathbf{x}$'s label or an embedding of $\mathbf{x}$. PAM can be used as a constituent component in a DNN to approximate a target $y = f(f_{PAM}(\mathbf{x}))$. For a classification task, we can calculate the conditional distribution $\hat{y} = Pr(y|\mathbf{x}) = \text{softmax}(f(f_{PAM}(\mathbf{x})))$ and optimize the cross-entropy $L_{CE} = -\mathbb{E}_{(\mathbf{x},y)\sim\mathcal{D}} y \log \hat{y} - (1-y)\log(1-\hat{y})$ between the observed $y$ and estimated $\hat{y}$ to determine the parameters in PAM. For a regression task, the mean square loss $L_{MSE} = \mathbb{E}_{(\mathbf{x},y)\sim\mathcal{D}}(y - f(f_{PAM}(\mathbf{x})))^2$ can be used.

## 4   Model Interpretation

We propose a conceptual framework to quantify and interpret the interaction effects learned by PAM. Without loss of generality, we assume that the DNN has a single output. Additional outputs can be similarly interpreted using the derived algorithm. The $f_{PAM}$ can be rewritten as a summation of $k$-way interaction terms for all possible values of $k \leq D$: $f_{PAM}(\mathbf{x}) = \sum_{\mathcal{I} \subseteq \{1,2,...,p\}} \phi_\mathcal{I}(\mathbf{x})$ where $\phi_\mathcal{I}(\mathbf{x})$ captures the total contribution to the output from the $|\mathcal{I}|$-way interactions among the features indexed in $\mathcal{I}$. If $\mathcal{I} \neq \varnothing$, $\phi_\mathcal{I}(\mathbf{x}) = \sum_{\sum m_i \leq D} w_\mathcal{I} \prod_{i \in \mathcal{I}} x_i^{m_i}$ where $x_i$ represents the $i^{th}$ feature of $\mathbf{x}$, $m_i$ calculates the power of $x_i$ in the interaction, and $w$'s are constant coefficients in front of the corresponding interaction terms. Given the definition of our attention in Eq.9, the highest polynomial order is $D - 1$ in the attention, together with the affine value function, the highest polynomial order of $f_{PAM}$ is $D$, so $\sum m_i$ can not exceed the depth of the tree. If $\mathcal{I} = \varnothing$, $\phi_\mathcal{I} = w_\varnothing$ which is a constant. We develop a method here to estimate the contribution values $\phi_\mathcal{I}$ in $f_{PAM} = \sum_{\mathcal{I} \subseteq \{1,2,...,p\}} \phi_\mathcal{I}$ for a fixed input $\mathbf{x}$ without computing the individual polynomial terms.

---

**Algorithm 1:** Obtain $\phi_\mathcal{I}$ for an input $\mathbf{x}$

1: **Input:** input $\mathbf{x}$, $g(\mathbf{x})$, and an $\mathcal{I} \subseteq \{1, 2, ..., p\}$
2: **Output:** $\phi_\mathcal{I}$
3: Set $\mathbf{0}^{-\mathcal{I}}$ to be a $p$-length vector with 1's in the positions indexed by $\mathcal{I}$ and 0's elsewhere
4: Calculate the following function: (Note that $\phi_{\mathcal{I}'}(\mathbf{x})$ is also calculated via the same recursive function.)

$$\phi_\mathcal{I}(\mathbf{x}) = \begin{cases} g(\mathbf{0}^{-\mathcal{I}} \odot \mathbf{x}) - \sum_{\mathcal{I}' \subset \mathcal{I}} \phi_{\mathcal{I}'}(\mathbf{x}), & \mathcal{I} \neq \varnothing, \\ g(\mathbf{0}), & \mathcal{I} = \varnothing, \end{cases}$$

where $\odot$ is the Hadamard product operator.

---

For a given $\mathbf{x}$, $f_{PAM}$ can be written explicitly out as $g(\mathbf{x})$ according to which polyhedron(s) $\mathbf{x}$ belongs to. To determine $g(\mathbf{x})$, we pass $\mathbf{x}$ through the model Eq.8 and evaluate every term in Eq.9. For instance, for the first product in Eq.9, if $\mathbf{x}$ makes $W_i\mathbf{x} + b_i \geq U_i$, we replace the term $\max(\min(W_i\mathbf{x} + b_i + U_i, 2U_i), 0)$ by $2U_i$; if $W_i\mathbf{x} + b_i \leq -U_i$, we replace it by 0; or otherwise, we use $W_i\mathbf{x} + b_i + U_i$. Once the $g$ function is computed, we can use Algorithm 1 to evaluate the contribution $\phi_\mathcal{I}, \forall \mathcal{I} \subseteq \{1, \cdots, p\}$.

A simple example can be used to demonstrate Algorithm 1. Let $\mathcal{I} = \{1, 2\}$ which means we calculate the sum of those cross terms that involve exactly $x_1$ and $x_2$. Thus we set all other elements in $\mathbf{x}$ to 0

and calculate $g(\mathbf{0}^{-\mathcal{I}} \odot \mathbf{x})$ to obtain the value $v$ that adds the terms involving only $x_1$ and $x_2$. We then additionally set either $x_1 = 0$ or $x_2 = 0$, or $x_1 = x_2 = 0$ (i.e., make all elements 0), re-compute $g$ to estimate the linear terms of either $x_1$ or $x_2$, or the constant term in $g$, and subtract these values from $v$ to eventually obtain $\phi_{\mathcal{I}}$. The following theorem characterize our results.

**Theorem 2** *For any input* $\mathbf{x}$, *by calculating* $\phi_{\mathcal{I}}(\mathbf{x})$ *for each* $\mathcal{I} \subseteq \{1, 2, ..., p\}$ *via Algorithm 1, we have* $\sum_{\mathcal{I} \subseteq \{1,2,...,p\}} \phi_{\mathcal{I}}(\mathbf{x}) = f_{PAM}(\mathbf{x})$.

The instance-level explanation of $f_{PAM}$ can be obtained by examining the magnitude of $\phi_{\mathcal{I}}$ which reflects the impact of the feature interaction among the features in $\mathcal{I}$. If $\phi_{\mathcal{I}}(\mathbf{x}) > 0$, the interaction increase the predicted value; if $\phi_{\mathcal{I}} < 0$, it reduces the output. The model-level interpretation can be approximated by computing the mean absolute value of the $\phi_{\mathcal{I}}$ across all sample instances.

## 5 Theoretical Justification - Approximation Theorems

We examine whether using PAM can enhance the expression power for universal approximation. We first introduce the Sobolev space, which characterizes a space of functions satisfying specific smoothness properties - Lipschitz continuous up to order $n$ - which is formally defined as:

**Definition 2 (Sobolev space)** *Let* $\mathcal{W}^{n,\infty}([0,1]^p)$ *be the Sobolev space which comprises of functions on* $[0,1]^p$ *lying in* $L^\infty$ *along with their weak derivatives up to order* $n$. *The norm of a function f in* $\mathcal{W}^{n,\infty}([0,1]^p)$ *is*

$$||f||_{\mathcal{W}^{n,\infty}([0,1]^p)} = \max_{\mathbf{n}:|\mathbf{n}| \leq n} ess \sup_{\mathbf{x} \in [0,1]^p} |D^n f(\mathbf{x})|,$$

*where* $\mathbf{n} = (n_1, n_2, ..., n_p) \in \{1, 2, ..., n\}^p$, $|\mathbf{n}| = n_1 + n_2 + ... + n_p \leq n$, *and* $D^{\mathbf{n}} f$ *is the* $\mathbf{n}$-*order weak derivative. The essential supreme* $ess \sup g(E) = \inf\{M \in R : \mu(\{x \in E : f(x) > M\}) = 0\}$ *captures the smallest value that the function* $g$ *can approach or exceed on a set* $E$, *except for a negligible subset of points with the measure* $\mu$. *Essentially, the space* $\mathcal{W}^{n,\infty}([0,1]^p)$ *is* $C^{n-1}([0,1]^p)$ *whose functions' derivatives up to order n are Lipschitz continuous.*

The following assumption is commonly used in the discussion of DNNs. Without loss of generality, it narrows our focus on a normalized Sobolev sphere. This assumption constrains the functions having Sobolev norm no greater than 1 within the sphere.

**Assumption 1** *Let* $F_{n,p}$ *be a set of functions lying in the unit ball in* $\mathcal{W}^{n,\infty}([0,1]^p)$, *we have*

$$F_{n,p} = \{f \in \mathcal{W}^{n,\infty}([0,1]^p) : ||f||_{\mathcal{W}^{n,\infty}}([0,1]^p) \leq 1\}.$$

This assumption is sufficient for our analysis, as functions encountered in real-world learning tasks can typically be linearly transformed into $\mathcal{W}^{n,\infty}([0,1]^p)$, as shown in previous studies [29]. This allows us to analyze the error bounds for terms in the polynomial approximation after performing Taylor expansion. Theorem 3 demonstrates that our model can almost surely approximate any ReLU-activated DNN model without error. All proofs can be found in Appendix E and F.

**Theorem 3** *If* $\mathbf{x}$ *is bounded and sampled from a distribution with upper-bounded probability density function, then for any ReLU activated plain DNN model* $f_{DNN}(\mathbf{x})$, *there exists a PAM with*

$$Pr(f_{PAM}(\mathbf{x}) = f_{DNN}(\mathbf{x})) \to 1.$$

Theorem 4 examines the parameter efficiency, and demonstrates that networks incorporating the proposed polyhedron attention require fewer parameters compared to those relying solely on ReLU activation, while maintaining the same approximation error in fitting functions in the Sobolev space.

**Theorem 4** *For any* $p$, $n > 0$ *and* $\epsilon \in (0, 1)$, *we have a PAM which can 1) approximates any function from* $F_{n,p}$ *with an error bound* $\epsilon$ *in the sense of* $L^\infty$ *with at most* $2p^n(N+1)^p(p+n-1)$ *parameters, where* $N = \lceil (\frac{n!}{2^p p^n} \frac{\epsilon}{2})^{-\frac{1}{n}} \rceil$.

**Remark 2** *For the purpose of comparison, the ReLU-activated plain DNN needs* $p^n(N+1)^p(p+1)n\mathcal{O}(\log(1/\epsilon))$ *parameters under the same setting in Theorem 4 [30, 29].*

It is worth noting that extensive research has been conducted on the approximation theory of DNNs with the ReLU-activation, which often concerns common function classes in specific function spaces such as Besov [31, 32] and Sobolev spaces [33, 34]. These analyses reveal a notable influence of the smoothness of the target function on the resulting approximation errors. Since Sobolev spaces characterize the smoothness properties of functions, we can investigate the ability of neural networks with ReLU activation to approximate functions with different degrees of regularity and smoothness. These theorems highlight the expressivity of our model and provide theoretical insights for the parameter efficiency of our proposed attention module in neural network architectures.

## 6 Empirical Evaluation

We evaluate the effectiveness and efficiency of PAM on three large-scale datasets: the **Criteo**[1] and **Avazu**[1] click-through-rate (CTR) datasets, and the **UK Biobank**[2] medical database. We conduct an analysis of the hyperparameters of PAM, and perform ablation studies by individually removing each of the three key components of PAM and evaluating the performance variations. Given the lack of known feature meanings in CTR benchmark datasets, we utilize the UK Biobank dataset as an example for studying model interpretation. Specifically, we validate the interaction effects captured by our interpretation framework, as detailed in Sec. 4, in the prediction of brain-age by the grey matter volumes from distinct brain regions. For implementation details, computation and space complexity of our model, please refer to Appendix G.

### 6.1 Experimental Setup

**Datasets.** Both the Criteo and the Avazu are massive industry datasets containing feature values and click feedback for display ads, and are processed following the benchmark protocol in BARS [35, 36]. The UK Biobank serves as a comprehensive biomedical database and research resource, offering extensive genetic and health-related information, where our objective is to predict participants' age by leveraging the grey matter volumes from 139 distinct brain regions. The summary statistics are listed in Table 1.

Table 1: Statistics of the datasets.

| Dataset | #Train | #Valid | #Test | #Features |
|---|---|---|---|---|
| Criteo | 33M | 8.3M | 4.6M | 2.1M |
| Avazu | 28.3M | 4M | 8.1M | 1.5M |
| UK Biobank | 31.8K | 4K | 4K | 139 |

**Evaluation metrics.** Criteo and Avazu datasets are concerned with binary classification tasks, evaluated using Area Under the ROC Curve (AUC). For brain-age prediction, a regression problem, we assess performance using the $R^2$ score.

**Baseline Algorithms.** We compare PAM against the top 5 algorithms from a pool of 33 baseline methods, selected based on the overall AUC scores in BARS, including DESTINE [19] (currently the best performer on Avazu), DCN [37], AOANet [4], EDCN [27] (best on Criteo), and DCN-V2 [25]. Given that our model and the selected algorithms are DNN-based, we also include a well-tuned DNN[1] as a strong baseline for our model comparison. It is worth noting that several state-of-the-art studies on CTR datasets, such as Criteo and Avazu, from Google [2, 24, 37], Microsoft [38] and Huawei [24], have recognized that even a marginal improvement of AUC at the level of 1‰ is considered a significant performance enhancement.

### 6.2 Effectiveness and efficiency evaluation

**Performance Comparison.** We compare the results over five runs. As shown in Fig. 4a, PAM achieves the highest AUC and $R^2$ score amongst all benchmark algorithms. PAM outperforms the second-best model by 1.4 ‰ for Criteo, 0.6 ‰ for Avazu, and 2.2 for UK Biobank, which is much higher than the improvement achieved by the second-best model from the next model (0.4 ‰ for Criteo, 0.16 ‰ for Avazu and 0.6 ‰ for UK Biobank). It is also worth noting that, despite being well-tuned, none of the benchmark algorithms performs optimally across all three datasets simultaneously. We evaluate the computation time of PAM by comparing the training and infereing runtime (seconds) per batch to the number of trainable parameters (#Par). As shown in Table 2, almost all models have a similar amount of free parameters, and their runtime per batch is comparable among the methods (PAM is superior than half of the methods and inferior than the other half.)

---

[1]https://github.com/openbenchmark/BARS/tree/master; [2]https://www.ukbiobank.ac.uk/

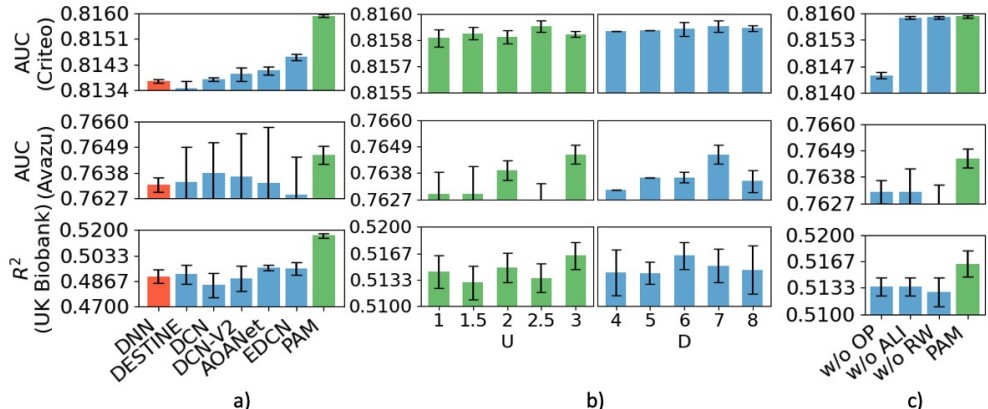

Figure 4: Experimental results. Error bars represent the means and standard deviations of AUC for classification and $R^2$ for regression. a) AUC and $R^2$ scores of PAM and comparison methods. b) Hyper-parameter analysis of PAM. U: the initial value of $U_n$. D: the depth of the oblique tree in PAM. c) Ablation studies of PAM.

**Hyper-parameter Study.** Although $U_i$'s are trainable parameters in PAM, their initialization may affect how well PAM works. We hence study the effects of the initial values of $U_i$'s by testing a set of choices from 1 to 3 by a step size of 0.5. We also examine the effects of the tree depth $D$ on PAM's performance. Fig. 4b shows the comparison where the initial values of $U_n$ do not sub-

Table 2: The number of parameters (#Par) and seconds per batch (Sec/Batch) during the training and inference of PAM and baseline models.

| Model | Criteo | | | Avazu | | | UK Biobank | | |
|---|---|---|---|---|---|---|---|---|---|
| | #Par | Sec/Batch | | #Par | Sec/Batch | | #Par | Sec/Batch | |
| | | Training | Interence | | Training | Interence | | Training | Interence |
| PAM | 22.3M | 0.102 | 0.0330 | 14.1M | 0.090 | 0.0263 | 315K | 0.083 | 0.0018 |
| DNN | 21.8M | 0.046 | 0.0139 | 13.8M | 0.038 | 0.0129 | 380K | 0.056 | 0.0063 |
| DESTINE | 21.5M | 0.130 | 0.0344 | 13.6M | 0.090 | 0.0188 | 384K | 0.072 | 0.0069 |
| DCN | 21.3M | 0.044 | 0.0114 | 13.4M | 0.080 | 0.0099 | 381K | 0.069 | 0.0064 |
| DCN-V2 | 22.0M | 0.103 | 0.0128 | 13.8M | 0.091 | 0.0137 | 458K | 0.059 | 0.0067 |
| AOANet | 21.4M | 0.151 | 0.0314 | 13.4M | 0.338 | 0.0168 | 457K | 0.066 | 0.0067 |
| EDCN | 21.5M | 0.066 | 0.0119 | 13.1M | 0.048 | 0.0113 | 63K | 0.072 | 0.0071 |

stantially change PAM's performance, which indicates that $U_n$ may be well trained during the PAM optimization. As for the tree depth $D$, the performance of PAM initially improves as $D$ increases, but when $D$ becomes large, the performance may deteriorate. An appropriate value of $D$ will help PAM to learn more domain knowledge, but if $D$ is too large, highly complex interactions may not necessarily lead to better prediction due to issues such as overfitting.

**Ablation Study of PAM.** The following specifies the three components for our ablation experiments.
**1) PAM without overlapping polyhedrons (PAM w/o OP).** As shown in Eqs. 5, 6 and 7, overlapping polyhedrons are generated in PAM to fit the target. To demonstrate the importance of soft splitting, we calculate $a_n(\mathbf{x})$ with $\prod_{i \in \mathcal{P}_n^l} \max(\min(W_i\mathbf{x} + b_i, 2U_i), 0) \prod_{i \in \mathcal{P}_n^r} \max(\min(-W_i\mathbf{x} - b_i, 2U_i), 0)$.
**2) PAM without adaptively learning interaction (PAM w/o ALI).** As shown in Fig. 5, the upper-bound $2U_i$ enables PAM to learn interactions with different orders in different polyhedrons. To exam the effectiveness of the upper bound, we remove this upper bound $2U_i$ from Eq. 9.
**3) PAM without removing $||W_i||$ (PAM w/o RW).** According to Theorem 1, the denominator of Eq. 7 can be removed without reducing the expression capability of PAM. To examine the correctness of this claim, we directly use PAM without removing $||W_i||$ from the denominator.
Fig. 4c clearly demonstrates a notable decline in the performance of the PAM when key components are removed. In particular, the standard PAM significantly outperforms the PAM w/o OP. It confirms the critical role of overlapping polyhedrons in enhancing PAM's performance. The removal of ALI decreases the AUC and $R^2$ of PAM as well, indicating the significance of combining low-way interaction effects with high-way ones to fit the target. Moreover, PAM w/o RW shows that removing $||W_i||$ from the denominator in Eq. 7 improves PAM's performance. Although, according to Theorem 1, PAM without RW has the same expressive capability as standard PAM, the inclusion of $W$'s norm in the denominator of Eq. 7 may result in unstable gradient variances, potentially compromising the performance of the optimizer.

### 6.3 Identified interaction effects among neural markers in predicting brain age

Although the interpretation framework was developed for PAM, it can be applied to those DNN architectures whose functions are piece-wise polynomials (see Appendix D). After carefully checking

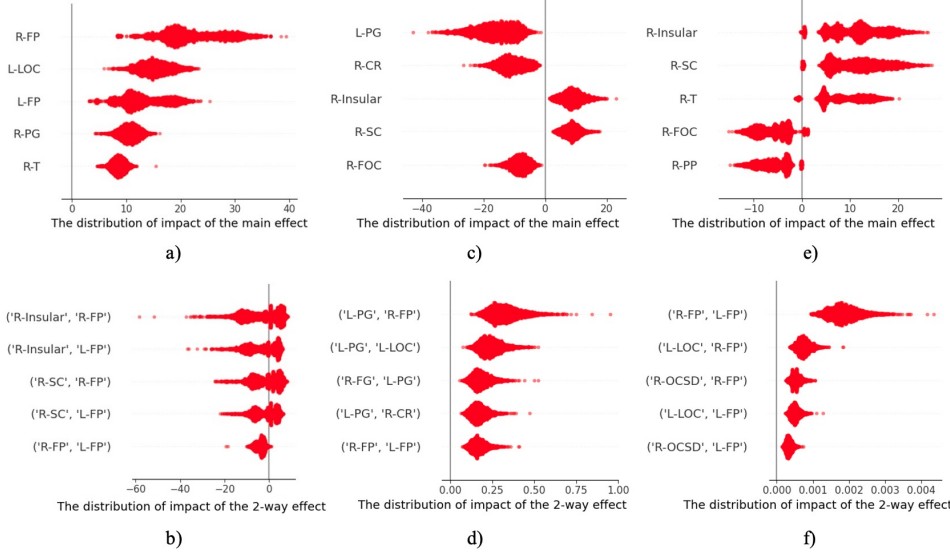

Figure 5: Top 5 main and 2-way interactions identified by PAM (Figs. a and b), DCN (Figs. c and d) and AOANet (Figs. e and f) by sorting mean absolute values of each $\phi_\mathcal{I}$ averaged over all participants. For each effect, points (each corresponding to a participant) are distributed according to $\phi$ values calculated by Algorithm 1 in the beeswarm bars. L: left; R: right; CR: crus II cerebellum; FOC: frontal orbital Cortex; FG: frontal gyrus; PP: planum polare; OCSD: lateral occipital cortex, superior division; FP: frontal pole; LOC: lateral occipital cortex; PG: precentral gyrus; T: thalamus; SC: subcallosal cortex.

all baseline methods, we found that our interpretation framework could be used to extract interactions from DCN and AOANet. Fig. 5 presents the top five main effects (Figs. a, c and e) and two-way interaction effects (Figs. b, d and f) between brain regions to brain-age prediction, as determined by the trained PAM, DCN and AOANet.

According to Figs. 5a and 5b, PAM found that the grey matter volumes (GMV) of both the left and right frontal poles (FP) play significant main effect as individual features, which is well aligned with previous research findings[39, 40, 41]. The other three brain regions, lateral occipital cortex (LOC), precentral gyrus (PG) and thalamus (T), are also discussed in early studies[42, 43, 44]. Existing aging studies primarily focus on main effects of GMV, but the top five two-way GMV interactions of identified brain regions have been acknowledged in works such as [42]. As shown in Fig. 5c, 5d, 5e and 5f, the main effects of PG and T identified by PAM were found by DCN and AOANet, respectively, and all three algorithms found the two-way interactions between the left and right FP. In addition to the shared top-5 brain regions identified by PAM, AOANet and DCN, PAM additionally identified the main effect of the LOC and FP and two-way interaction effects related to the insular and subcallosal cortex (SC) regions.

# 7   Conclusions and Future Work

We propose a novel feature interaction learning module, namely Polyhedron Attention Module (PAM), to fit a target with adaptive-order interaction effects. PAM produces a self-attention mechanism partitioning the input space into overlapping polyhedrons and learning the boundary hyperplanes for each polyhedron automatically. These hyperplanes multiply to identify interaction effects specific to individual polyhedrons. Under this mechanism, PAM automatically captures both simple and complex interaction effects. In our theoretic analysis, we show that PAM can enhance the model expression power for universal approximation. Experimental results demonstrate that the proposed model achieves better performance on massive datasets than the state-of-the-art. In the future, we will study how to dynamically pruning PAM's oblique tree during the training process to regularize the model. It will also be interesting to investigate how to define boundaries with non-linear functions.

## Acknowledgments

This research has been conducted using the UK Biobank Resource under the Application 51296 – Machine Learning Analysis of Brain Images, Behaviors, and Genotypes to Understand Mental Disorders, and was supported in part by the NIH grant K02-DA043063 and NSF award #2122309.

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
