# Polyhedron Attention Module: Learning Adaptive-order Interactions

# 1 Appendixes

## Contents

## A    Deriving Eq. 2.

We consider a $L$-layer ($L \geq 2$) ReLU activated plain DNN module $f : \mathbb{R}^{n_0} \to \mathbb{R}^{n_L}$ with input $\mathbf{x} \in \mathbb{R}^p$. Let $W^\ell \in \mathbb{R}^{n_\ell \times n_{\ell-1}}$ and $b^\ell \in \mathbb{R}^{n_\ell}$ be the weights and offset vectors of layer $\ell$, for $\ell = \{1, ..., L\}$ and $n_0 = p$. Let $f^0(\mathbf{x}) = \mathbf{x}$. For $\ell \in \{1, ..., L\}$, we define recursively the pre- and post-activation output of every layer as

$$g^\ell(\mathbf{x}) = W^\ell f^{\ell-1}(\mathbf{x}) + b^\ell,$$
$$f^\ell(\mathbf{x}) = \text{ReLU}(g^\ell(\mathbf{x})),$$

where ReLU activation function is denoted by $\text{ReLU}(t) = max(0, t)$.

The first $L-1$ layers of the ReLU-activated DNN module has $\sum_{\ell=1}^{L-1} n_\ell$ activation functions, and thus have $2^{\sum_{\ell=1}^{L-1} n_\ell}$ possible activation states. Let $A \in \{1, -1\}^{\sum_{\ell=1}^{L-1} n_\ell}$ be an activation state ($1/-1$ means activate/inactive) of all but the last layer's ReLU in the DNN module, and $A_i^\ell \in \{-1, 1\}$ be the activation state of the $i^{th}$ ReLU activation function in the $l^{th}$ layer of the DNN module. Conditioned on $\{A^1, ..., A^{\ell-1}\}$, $g^\ell(\mathbf{x})$ can be rewritten as a linear function.

$$
\begin{aligned}
&g^\ell(\mathbf{x})|_{\{A^1,...,A^{\ell-1}\}} \\
&= \begin{cases}
W_\varnothing^{(1)}\mathbf{x} + b_\varnothing^{(1)} \\
= W^1\mathbf{x} + b^1, & \ell = 1, \\
W_{\{A^1,...,A^{\ell-1}\}}^{(\ell)}\mathbf{x} + b_{\{A^1,...,A^{\ell-1}\}}^{(\ell)} \\
= W^\ell \Sigma^{A^{\ell-1}} W_{\{A^1,...,A^{\ell-2}\}}^{(\ell-1)}\mathbf{x} + W^\ell \Sigma^{A^{\ell-1}} b_{\{A^1,...,A^{\ell-2}\}}^{(\ell-1)} + b^\ell, & \ell > 1,
\end{cases}
\end{aligned}
\tag{1}
$$

where $W_{\{A^1,...,A^{\ell-1}\}}^{(\ell)} \in \mathbb{R}^{n_\ell \times n_0}$, $b_{\{A^1,...,A^{\ell-1}\}}^{(\ell)} \in \mathbb{R}^{n_\ell}$, $\Sigma^{A^\ell}$ is a $n_\ell \times n_\ell$ matrix with

$$\Sigma_{i,j}^{A^\ell} = \mathbb{1}(i = j \text{ and } A_i^\ell = 1).$$

To generate the activation state $A$, $\mathbf{x}$ should meet all inequalities $A_i^\ell g_i^\ell(\mathbf{x})|_{\{A^1,...,A^{\ell-1}\}} \geq 0$ for $\forall \ell \in \{1, 2, ..., L-1\}$ and $\forall i \in \{1, 2, ..., n_\ell\}$, where $g_i^\ell(\mathbf{x})|_{\{A^1,...,A^{\ell-1}\}}$ is the $i^{th}$ result of $g^\ell(\mathbf{x})|_{\{A^1,...,A^{\ell-1}\}}$. It results in a polyhedron $\Delta_A$:

$$\Delta_A = \cap_{\ell \in \{1,...,L-1\}} \cap_{i \in \{1,...,n_\ell\}} \{\mathbf{z} \in \mathbb{R}^p | A_i^\ell g_i^\ell(\mathbf{z})|_{\{A^1,...,A^{\ell-1}\}} \geq 0\}. \tag{2}$$

Let the set of polyhedron be $\mathcal{S}_\Delta = \{\Delta_A | A \in \{1, -1\}^{\sum_{l=1}^{L-1} n_\ell}\}$. We have

$$
g^L(\mathbf{x}) = \begin{cases}
W_{\Delta_1}^{(L)}\mathbf{x} + b_{\Delta_1}^{(L)}, & \mathbf{x} \in \Delta_1, \\
... \\
W_{\Delta_{|\mathcal{S}_\Delta|}}^{(L)}\mathbf{x} + b_{\Delta_{|\mathcal{S}_\Delta|}}^{(L)}, & \mathbf{x} \in \Delta_{|\mathcal{S}_\Delta|}.
\end{cases}
\tag{3}
$$

Then the $i^{th}$ activation function's output in the last layer of the DNN can be expressed as

$$
ReLU(g_i^L(\mathbf{x})) = \sum_{\Delta \in \mathcal{S}_\Delta} \mathbb{1}(\mathbf{x} \in \Delta) ReLU(W_{\Delta,i}^{(L)}\mathbf{x} + b_{\Delta,i}^{(L)})
$$

$$
= \sum_{\Delta \in \mathcal{S}_\Delta} \mathbb{1}(\mathbf{x} \in \Delta, W_{\Delta,i}^{(L)}\mathbf{x} + b_{\Delta,i}^{(L)} \geq 0)(W_{\Delta,i}^{(L)}\mathbf{x} + b_{\Delta,i}^{(L)}) + \sum_{\Delta \in \mathcal{S}_\Delta} \mathbb{1}(\mathbf{x} \in \Delta, W_{\Delta,i}^{(L)}\mathbf{x} + b_{\Delta,i}^{(L)} < 0) \cdot 0
$$

$$
= \sum_{\Delta \in \mathcal{S}_\Delta} \mathbb{1}(\mathbf{x} \in \Delta, W_{\Delta,i}^{(L)}\mathbf{x} + b_{\Delta,i}^{(L)} \geq 0) dist(\mathbf{x}, H_{\Delta,i,L}) ||W_{\Delta,i}^{(L)}||
$$

$$
+ \sum_{\Delta \in \mathcal{S}_\Delta} \mathbb{1}(\mathbf{x} \in \Delta, W_{\Delta,i}^{(L)}\mathbf{x} + b_{\Delta,i}^{(L)} < 0) dist(\mathbf{x}, H_{\Delta,i,L}) \cdot 0,
$$

where $g_i^L(\mathbf{x})$ is the $i^{th}$ element of $g^L(\mathbf{x})$'s output, $W_{\Delta,i}^{(L)}$ is the $i^{th}$ row of $W_\Delta^{(L)}$, $b_{\Delta,i}^{(L)}$ is the $i^{th}$ element of $b_\Delta^{(L)}$, and $dist(\mathbf{x}, H_{\Delta,i,L})$ is the distance from $\mathbf{x}$ to $W_{\Delta,i}^{(L)}\mathbf{x} + b_{\Delta,i}^{(L)} = 0$. Let $\Delta_1 = \{\mathbf{x} \in \Delta, W_{\Delta,i}^{(L)}\mathbf{x} + b_{\Delta,i}^{(L)} \geq 0\}$ and $\Delta_2 = \{\mathbf{x} \in \Delta, W_{\Delta,i}^{(L)}\mathbf{x} + b_{\Delta,i}^{(L)} < 0\}$. Eq. 2 in the main text can be obtained by rewriting $\sum_{\Delta \in \mathcal{S}_\Delta}$ as $\sum_\Delta$.

## B   The hyperplane set generated by the oblique tree is a superset of that created by the ReLU-activated plain DNN

An oblique tree is a binary tree where each node splits the space by a hyperplane rather than by thresholding a single feature. The tree starts with the root of the full input space $\mathcal{S}$, and by recursively splitting $\mathcal{S}$, the tree grows deeper. For a $D$-depth ($D \geq 3$) binary tree, there are $2^{D-1} - 1$ internal nodes and $2^{D-1}$ leaf nodes. As shown in Fig. 3, each internal and leaf node maintains a sub-space representing a polyhedron $\Delta$ in $\mathcal{S}$, and each layer of the tree corresponds to a partition of the input space into polyhedrons. Denote the polyhedron defined in node $n$ by $\Delta_n$, and the left and right child nodes of $n$ by $n_L$ and $n_R$. We perform soft partition to split each $\Delta_n$ into $\Delta_{n_L}$ and $\Delta_{n_R}$ with an overlapping buffer. Let the splitting hyperplane be $\{\mathbf{x} \in \mathbb{R}^p : W_n \mathbf{x} + b_n = 0\}$. Then the two sub-spaces $\Delta_{n_L}$ and $\Delta_{n_R}$ are defined as follows:

$$\begin{aligned} \Delta_{n_L} &= \{\mathbf{x} \in \Delta_n | \, W_n \mathbf{x} + b_n \geq -U_n\}, \\ \Delta_{n_R} &= \{\mathbf{x} \in \Delta_n | \, -W_n \mathbf{x} - b_n \geq -U_n\}, \end{aligned} \tag{4}$$

where $U_n$ indicates the width of the overlapping buffer.

According to the Appendix A, a ReLU-activated plain DNN $g^L(\mathbf{x})$ can be rewritten as a piece-wise linear function dividing the input space into a set of polyhedrons $\mathcal{S}_\Delta$. In this section, we are going to prove that for any $\mathcal{S}_\Delta$ generated by the ReLU-activated plain DNN, there exists an oblique tree dividing the input space into the same polyhedron set.

**Proof:   Statement**: For any $\mathcal{S}_\Delta$ generated by the ReLU-activated plain DNN, there exists an oblique tree dividing the input space into the same polyhedron set.

**Base Case**: Let $A^\ell = \{1, -1\}^{n_\ell}$ be the activation state ($1/-1$ means activate/ inactive) of the $\ell^{th}$ layer of the DNN. To generate the activation state $A^1$, the input of the ReLU-activated plain DNN $\mathbf{x}$ belongs to the polyhedron

$$\Delta_{\{A^1\}} = \cap_{i \in \{1,\dots,n_1\}} \{\mathbf{z} \in \mathbb{R}^p | A_i^1 g_i^1(\mathbf{z})|_\varnothing \geq 0\},$$

where $g_i^1(\mathbf{x})|_\varnothing$ is defined in Eq. 1. We can build an oblique tree $T^1$ to generate $\Delta_{\{A^1\}}$. In particular, the depth of the oblique tree is $n_1 + 1$. Let $\mathcal{N}_d$ be the oblique tree's node set with depth $d$ ($d \in \{1, 2, \dots, n_1\}$). For each node $n \in \mathcal{N}_d$, we have $W_n = W_d^1$, $b_n = b_d^1$, and $U_n = 0$ (see definitions in Eq. 4). According to Eq. 6 in the main text, for each $T^1$'s leaf node $n$, the oblique tree generates polyhedrons following

$$\Delta_n = \left[ \cap_{n' \in \mathcal{P}_n^l} \{\mathbf{z} \in \mathbb{R}^p | W_{n'} \mathbf{z} + b_{n'} \geq 0\} \right] \cap \left[ \cap_{n' \in \mathcal{P}_n^r} \{\mathbf{z} \in \mathbb{R}^p | (-W_{n'}) \mathbf{z} + (-b_{n'}) \geq 0\} \right].$$

For any possible $A^1$, we can find a leaf node $n$ from $\mathcal{N}_{n_1+1}$ with $\Delta_n = \Delta_{\{A^1\}}$. Then for each leaf node $n$, we can also find a activation state with $\Delta_{\{A^1\}} = \Delta_n$. Therefore, we have $\{\Delta_{\{A^1\}} | A^{\ell'} \in \{1, -1\}^{n_{\ell'}}, \ell' \in \{1\}\} = \{\Delta_n | n \text{ is } T^1\text{'s leaf node}\}$.

**Inductive Step**: To generate activation states $\{A^1, \dots, A^{\ell-1}\}$ ($\ell > 1$), according to Eq. 2, the input of the DNN model belongs to

$$\Delta_{\{A^1,\dots,A^{\ell-1}\}} = \cap_{\ell' \in \{1,\dots,\ell-1\}} \cap_{i \in \{1,\dots,n_{\ell'}\}} \{\mathbf{z} \in \mathbb{R}^p | A_i^{\ell'} g_i^{\ell'}(\mathbf{z})|_{\{A^1,\dots,A^{\ell'-1}\}} \geq 0\}.$$

If there exists a $(\sum_{l'=1}^{l-1} n_{\ell'} + 1)$-depth oblique tree $T^{\ell-1}$ splitting the input space into a set of polyhedrons with $\{\Delta_{\{A^1,\dots,A^{\ell-1}\}} | A^{\ell'} \in \{1, -1\}^{n_{\ell'}}, \ell' \in \{1, 2, \dots, \ell-1\}\} = \{\Delta_n | n \text{ is } T^{\ell-1}\text{'s leaf node}\}$, by adding nodes to $T^{\ell-1}$, we could build a $(\sum_{l'=1}^{l} n_{\ell'} + 1)$-depth oblique tree $T^\ell$ with $\{\Delta_{\{A^1,\dots,A^\ell\}} | A^{\ell'} \in \{1, -1\}^{n_{\ell'}}, \ell' \in \{1, 2, \dots, \ell\}\} = \{\Delta_n | n \text{ is } T^\ell\text{'s leaf node}\}$. In the following part, we exhibit the pipeline to build $T^\ell$.

For each $T^{\ell-1}$'s leaf node $n$ with $\Delta_n = \Delta_{\{A^1,\dots,A^{\ell-1}\}}$, We build an oblique sub-tree rooted at $n$ to generate

$$\Delta_{\{A^1,\dots,A^\ell\}} = \Delta_{\{A^1,\dots,A^{\ell-1}\}} \cap \left[ \cap_{i \in \{1,\dots,n_\ell\}} \{\mathbf{z} \in \mathbb{R}^p | A_i^\ell g_i^\ell(\mathbf{z})|_{\{A^1,\dots,A^{\ell-1}\}} \geq 0\} \right].$$

In particular, the depth of the oblique sub-tree is $n_\ell + 1$. Let $\mathcal{N}_d$ be the oblique sub-tree's node set with depth $d$. For each node $n \in \mathcal{N}_d$, we have $W_n = W_{\{A^1,\dots,A^{\ell-1}\},d}^{(\ell)}$, $b_n = b_{\{A^1,\dots,A^{\ell-1}\},d}^{(\ell)}$, and

72   $U_n = 0$. After adding sub-trees to each of $T^{\ell-1}$'s leaf nodes to form $T^\ell$, for any activation state
73   $\{A^1, ..., A^\ell\}$, we can find a leaf node $n \in \mathcal{N}_{\sum_{l'=1}^{l} n^{l'}+1}$ from $T^\ell$ with $\Delta_n = \Delta_{\{A^1,...,A^\ell\}}$.

74   **Conclusion**: According to the base case and the inductive step, for any ReLU-activated plain DNN's
75   $\mathcal{S}_\Delta = \{\Delta_{\{A^1,...,A^{L-1}\}}|A^\ell \in \{1,-1\}^{n_\ell}, \ell \in \{1,2,...,L-1\}\}$, we can build an oblique tree $T^{L-1}$
76   with $\mathcal{S}_\Delta = \{\Delta_n|n$ is $T^{L-1}$'s leaf nodes$\}$.       $\square$

## 77   C   Proof of Theorem 1

78   If all value functions $V$ belong to a function set that is closed under linear transformations, then the
79   function learned by PAM $f_{PAM}$ can be equivalently written as

$$f_{PAM}(\mathbf{x}) = V(\mathbf{x}, \theta_G) + \sum_{n \in \mathcal{S}_\Delta^-} a_n(\mathbf{x})V(\mathbf{x}, \theta_n) \tag{5}$$

80   where the polyhedron set $\mathcal{S}_\Delta^-$ contains half of the polyhedrons (e.g., the right child nodes or the left
81   child nodes) in $\mathcal{S}_\Delta$ and

$$a_n(\mathbf{x}) = \prod_{i \in \mathcal{P}_n^l} \max(\min(W_i\mathbf{x}+b_i+U_i, 2U_i), 0) \prod_{i \in \mathcal{P}_n^r} \max(\min(-W_i\mathbf{x}-b_i+U_i, 2U_i), 0). \tag{6}$$

82   **Proof:**   Suppose that both $V(\mathbf{x}, \theta_G)$ and $V(\mathbf{x}, \theta_n)$ in Eq. 5 belong to the function set $\mathcal{V}$. Then we
83   have

$$a_n(\mathbf{x})V(\mathbf{x}, \theta_n)$$
$$= \prod_{i \in \mathcal{P}_n^l} \max(\min(\frac{W_i\mathbf{x}+b_i+U_i}{||W_i||}, \frac{2U_i}{||W_i||}), 0) \prod_{i \in \mathcal{P}_n^r} \max(\min(\frac{-W_i\mathbf{x}-b_i+U_i}{||W_i||}, \frac{2U_i}{||W_i||}), 0)$$
$$\times \left[V(\mathbf{x}, \theta_n) \prod_{i \in \mathcal{P}_n^l} ||W_i|| \prod_{i \in \mathcal{P}_n^r} ||W_i||\right]$$

84   Since $\mathcal{V}$ is a function set closed under linear transformation, there exists a value function $V(\mathbf{x}, \theta_n') =$
85   $\left[V(\mathbf{x}, \theta_n) \prod_{i \in \mathcal{P}_n^l} ||W_i|| \prod_{i \in \mathcal{P}_n^r} ||W_i||\right]$ with $V(\mathbf{x}, \theta_n') \in \mathcal{V}$. Therefore, removing the 2-norm of $W_i$
86   will not decrease the expression capability of attention.

87   To prove that $\mathcal{S}_\Delta$ can be replaced with $\mathcal{S}_\Delta^-$, with $\mathcal{S}_\Delta = \cup_{d=2}^{D} \{\Delta_n | n \in \mathcal{N}_d\}$, we rewrite the output of
88   PAM (Eq. 4 in the main text) to

$$f_{PAM}(\mathbf{x}) = V(\mathbf{x}; \theta_G) + \sum_{d=2}^{D} \sum_{n \in \mathcal{N}_d} a_n(\mathbf{x})V(\mathbf{x}; \theta_n).$$

89   Then we start from depth $D$ to 2 and show that $\frac{|\mathcal{N}_d|}{2}$ value functions can be removed in each depth $d$.
90   First, let $P_n$ and $S_n$ be the parent and sibling nodes of $n$. We have

$$\sum_{n \in \mathcal{N}_D} a_n(\mathbf{x})V(\mathbf{x}; \theta_n) + \sum_{n \in \mathcal{N}_{D-1}} a_n(\mathbf{x})V(\mathbf{x}; \theta_n)$$
$$= \sum_{n \in \mathcal{N}_D} \mathbb{1}(n \text{ is the left child node})a_n(\mathbf{x})V(\mathbf{x}; \theta_n) + \sum_{n \in \mathcal{N}_D} \mathbb{1}(n \text{ is the right child node})a_n(\mathbf{x})V(\mathbf{x}; \theta_n)$$
$$+ \sum_{n \in \mathcal{N}_{D-1}} a_n(\mathbf{x})V(\mathbf{x}; \theta_n)$$
$$= \sum_{n \in \mathcal{N}_D} \mathbb{1}(n \text{ is the left child node})a_n(\mathbf{x})(V(\mathbf{x}; \theta_n) - V(\mathbf{x}; \theta_{S_n}))$$
$$+ \sum_{n \in \mathcal{N}_{D-1}} a_n(\mathbf{x})(2U_nV(\mathbf{x}; \theta_{n_R}) + V(\mathbf{x}; \theta_n))$$

$$\tag{7}$$

91   Since $V$ belongs to $\mathcal{V}$, a function set closed under linear transformation, we have

$$V(\mathbf{x}; \theta_n') = V(\mathbf{x}; \theta_n) - V(\mathbf{x}; \theta_{S_n}), \qquad n \in \mathcal{N}_D,$$
$$V(\mathbf{x}; \theta_n'') = 2U_nV(\mathbf{x}; \theta_{n_R}) + V_n(\mathbf{x}; \theta_n), \quad n \in \mathcal{N}_{D-1}.$$

92   Therefore, $\frac{|\mathcal{N}_D|}{2}$ attention scores and value functions can be removed in depth $D$. By replacing value
93   functions from depth D to 2 following Eq. 7, the number of value functions is halved in each depth of
94   the tree. It means that $\mathcal{S}_\Delta$ can be replaced by $\mathcal{S}_\Delta^-$.       $\square$

  **D   Proof of Theorem 2**

For any input $\mathbf{x}$, by calculating $\phi_{\mathcal{I}}(\mathbf{x})$ for each $\mathcal{I} \subseteq \{1, 2, ..., p\}$ via Algorithm 1, we have $\sum_{\mathcal{I} \subseteq \{1,2,...,p\}} \phi_{\mathcal{I}}(\mathbf{x}) = f_{PAM}(\mathbf{x})$.

**Proof:**   We first shows that $f_{PAM}$ can be written explicitly out as $g(\mathbf{x})$ according to which polyhedron(s) $\mathbf{x}$ belongs to.

As shown in Eq. 8, max and min operators in PAM's attentions can be rewritten as the ReLU-activated function

$$\max(\mathbf{z}, 0) = ReLU(\mathbf{z}) \ and \ \min(\mathbf{z}, 2U_i) = -ReLU(-\mathbf{z} + 2U_i) + 2U_i, \tag{8}$$

the calculation of each PAM's attention score contains 2 ReLU activation functions. Suppose that the $f_{PAM}$ has $n_a$ ReLU activation functions in total, which results in $2^{n_a}$ possible activation states. Let $A = \{A_1, ..., A_{n_a}\} \in \{1, -1\}^{n_a}$ be an activation state ($1/-1$ means activate/inactive) of $f_{PAM}(\mathbf{x})$, and $\mathcal{A}$ be the set containing all possible activation states. Let $ReLU(h_i(\mathbf{x}))$ be the $i^{th}$ ReLU activation function in $f_{PAM}(\mathbf{x})$. We have

$$f_{PAM}(\mathbf{x}) = \sum_{A \in \mathcal{A}} \Big[ \prod_{i=1}^{n_a} \mathbb{1}(A_i h_i(\mathbf{x}) \geq 0) \Big] g_A(\mathbf{x}), \tag{9}$$

where $g_A(\mathbf{x})$ is a polynomial function differentiable everywhere under the activation state $A$. In particular, if we have an activation state $A$, we can obtain $g_A(\mathbf{x})$ by replacing the ReLU activations in $f_{PAM}(\mathbf{x})$ with either $h_i(\mathbf{x})$ or 0 depending on whether the corresponding pre-activation value $h_i(\mathbf{x})$ is non-negative or negative, respectively. For the sake of simplicity, we simplify $g_A(\mathbf{x})$ as $g(\mathbf{x})$. Given the definition of our attention in Eq. 9 in the main text, the highest polynomial order is $D - 1$ in the attention, together with the affine value function, the highest polynomial order of $g(\mathbf{x})$ is $D$. Since we have assumed $f_{PAM}$ has only one output at the beginning of section 4 in the main text, $g(\mathbf{x}) : \mathbb{R}^p \to \mathbb{R}$ is a $D + 1$ times continuously differentiable function at every point $\mathbf{a} \in \mathbb{R}^p$, and $g(\mathbf{x})$'s $(D + 1)$-order partial derivatives always equals zero, the $D$ order Taylor polynomial of $g(\mathbf{x})$ at the point $\mathbf{a}$ is

$$g(\mathbf{x}) = \sum_{|\mathbf{m}| \leq D} \frac{D^{\mathbf{m}} g(\mathbf{a})}{\mathbf{m}!} (\mathbf{x} - \mathbf{a})^{\mathbf{m}} = \sum_{|\mathbf{m}| \leq D} w_{\mathbf{m}} \mathbf{x}^{\mathbf{m}}, \tag{10}$$

where $\mathbf{m} = \{m_1, m_2, ..., m_p\}$ with $m_i \in \mathbb{Z}^+$, $|\mathbf{m}| = m_1 + ... + m_p$, $\mathbf{m}! = m_1!...m_p!$, $\mathbf{x}^{\mathbf{m}} = x_1^{m_1}...x_p^{m_p}$, $D^{\mathbf{m}} g = \frac{\partial^{|\mathbf{m}|} g}{\partial x_1^{m_1}...\partial x_p^{m_p}}$ and $w_{\mathbf{m}} \in \mathbb{R}$ is the weight for the interaction term $\mathbf{x}^{\mathbf{m}}$. Let $\mathcal{I} \subseteq \{1, 2, ..., p\}$ be a set of $\mathbf{x}$'s feature indices. The interaction effects among $\mathbf{x}$'s elements indexed by $\mathcal{I}$ are defined by

$$\phi_{\mathcal{I}}(\mathbf{x}) = \sum_{|\mathbf{m}| \leq D} \mathbb{1}(\prod_{i \in \mathcal{I}} m_i > 0 \ and \sum_{i \in \{1,2,...,p\}/\mathcal{I}} m_i = 0) w_{\mathbf{m}} \mathbf{x}_1^{m_1}...x_p^{m_p},$$
$$\phi_{\varnothing}(\mathbf{x}) = g(\mathbf{0}) = w_{\mathbf{m}}|_{|\mathbf{m}|=0}, \tag{11}$$

where $\phi_{\varnothing}(\mathbf{x})$ is the constant effects. Obviously, we have

$$\sum_{\mathcal{I} \subseteq \{1,2,...,p\}} \phi_{\mathcal{I}}(\mathbf{x}) = g(\mathbf{x}).$$

122   The indicator function in Eq. 11 can be removed by rewriting $\phi_{\mathcal{I}}(\mathbf{x})$ as

$$
\begin{aligned}
\phi_{\mathcal{I}}(\mathbf{x}) &= \sum_{|\mathbf{m}|\leq D} \mathbb{1}(\prod_{i\in\mathcal{I}} m_i > 0)w_{\mathbf{m}}(\mathbf{0}^{-\mathcal{I}}\odot\mathbf{x})^{\mathbf{m}} \\
&= \sum_{|\mathbf{m}|\leq D} w_{\mathbf{m}}(\mathbf{0}^{-\mathcal{I}}\odot\mathbf{x})^{\mathbf{m}} - \sum_{|\mathbf{m}|\leq D} \mathbb{1}(\prod_{i\in\mathcal{I}} m_i = 0)w_{\mathbf{m}}(\mathbf{0}^{-\mathcal{I}}\odot\mathbf{x})^{\mathbf{m}} \\
&= \sum_{|\mathbf{m}|\leq D} w_{\mathbf{m}}(\mathbf{0}^{-\mathcal{I}}\odot\mathbf{x})^{\mathbf{m}} - \sum_{|\mathbf{m}|\leq D}\Big[\sum_{\mathcal{I}'\subset\mathcal{I}} \mathbb{1}(\prod_{i\in\mathcal{I}'} m_i > 0 \text{ and } \sum_{i\in\{1,2,\ldots,p\}/\mathcal{I}'} m_i = 0)\Big]w_{\mathbf{m}}(\mathbf{0}^{-\mathcal{I}}\odot\mathbf{x})^{\mathbf{m}} \\
&= \sum_{|\mathbf{m}|\leq D} w_{\mathbf{m}}(\mathbf{0}^{-\mathcal{I}}\odot\mathbf{x})^{\mathbf{m}} - \sum_{|\mathbf{m}|\leq D}\sum_{\mathcal{I}'\subset\mathcal{I}} \mathbb{1}(\prod_{i\in\mathcal{I}'} m_i > 0)w_{\mathbf{m}}(\mathbf{0}^{-\mathcal{I}'}\odot\mathbf{0}^{-\mathcal{I}}\odot\mathbf{x})^{\mathbf{m}} \\
&= \sum_{|\mathbf{m}|\leq D} w_{\mathbf{m}}(\mathbf{0}^{-\mathcal{I}}\odot\mathbf{x})^{\mathbf{m}} - \sum_{|\mathbf{m}|\leq D}\sum_{\mathcal{I}'\subset\mathcal{I}} \mathbb{1}(\prod_{i\in\mathcal{I}'} m_i > 0)w_{\mathbf{m}}(\mathbf{0}^{-\mathcal{I}'}\odot\mathbf{x})^{\mathbf{m}} \\
&= g(\mathbf{0}^{-\mathcal{I}}\odot\mathbf{x}) - \sum_{\mathcal{I}'\subset\mathcal{I}} \phi_{\mathcal{I}'}(\mathbf{x}).
\end{aligned}
$$

(12)

123   where $\mathbf{0}^{-\mathcal{I}}$ is a $d$-length zero vector with ones indexed by $\mathcal{I}$, $\odot$ is the Hadamard product operator.
124   Since we have $\phi_{\varnothing}(\mathbf{x}) = g(\mathbf{0})$, we can calcualte any $\phi_{\mathcal{I}}(\mathbf{x})$ by recursively calculating $\phi_{\mathcal{I}'}(\mathbf{x})$ for
125   every $\mathcal{I}$'s subset $\mathcal{I}'$.       $\square$

126 ## E   Proof of Theorem 3

127   If $\mathbf{x}$ is bounded and sampled from a distribution with upper-bounded probability density function,
128   then for any ReLU activated plain DNN model $f_{\text{DNN}}(\mathbf{x})$, there exists a PAM with

$$
Pr(f_{PAM}(\mathbf{x}) = f_{\text{DNN}}(\mathbf{x})) \to 1.
$$

129   **Proof:**   For any oblique tree's internal node $n$ in the PAM, we set $V(\mathbf{x};\theta_n)\equiv 0$. Then by setting
130   $V(\mathbf{x},\theta_G)\equiv 0$, with the set of $T$'s leaf node $\mathcal{N}_D$, we have

$$
\begin{aligned}
&f_{PAM}(\mathbf{x}) \\
&= \sum_{n'\in\mathcal{N}_D} a_{n'}(\mathbf{x})V(\mathbf{x};\theta_{n'}) \\
&= \sum_{n'\in\mathcal{N}_D} \prod_{i\in\mathcal{P}^l_{n'}} \max(\min(W_i\mathbf{x}+b_i+U_i, 2U_i), 0) \prod_{i\in\mathcal{P}^r_{n'}} \max(\min(-W_i\mathbf{x}-b_i+U_i, 2U_i), 0)V(\mathbf{x};\theta_{n'}) \\
&= \sum_{n'\in\mathcal{N}_D} \Big[\prod_{i\in\mathcal{P}^l_{n'}} \mathbb{1}(W_i\mathbf{x}+b_i+U_i \geq 2U_i) \prod_{i\in\mathcal{P}^r_{n'}} \mathbb{1}(-W_i\mathbf{x}-b_i+U_i \geq 2U_i)\big[\prod_{i\in\mathcal{P}^l_{n'}\cup\mathcal{P}^r_{n'}} 2U_i\big]V(\mathbf{x};\theta_{n'}) \\
&\quad +(1-\prod_{i\in\mathcal{P}^l_{n'}} \mathbb{1}(W_i\mathbf{x}+b_i+U_i \geq 2U_i) \prod_{i\in\mathcal{P}^r_{n'}} \mathbb{1}(-W_i\mathbf{x}-b_i+U_i \geq 2U_i))a_{n'}(\mathbf{x})V(\mathbf{x};\theta_{n'})\Big].
\end{aligned}
$$

(13)

131   Given $\mathbf{x}$, if there exists a $T$'s leaf node $n'$ with

$$
\prod_{i\in\mathcal{P}^l_{n'}} \mathbb{1}(W_i\mathbf{x}+b_i+U_i \geq 2U_i) \prod_{i\in\mathcal{P}^r_{n'}} \mathbb{1}(-W_i\mathbf{x}-b_i+U_i \geq 2U_i) = 1,
$$

132   we have

$$
\prod_{i\in\mathcal{P}^l_{n'}} \mathbb{1}(W_i\mathbf{x}+b_i \geq 0) \prod_{i\in\mathcal{P}^r_{n'}} \mathbb{1}(-W_i\mathbf{x}-b_i \geq 0) = 1,
$$

(14)

133   Therefore, conditioned on the event

$$
\mathcal{E} = \{\exists n'\in\mathcal{N}_D, \prod_{i\in\mathcal{P}^l_{n'}} \mathbb{1}(W_i\mathbf{x}+b_i+U_i \geq 2U_i) \prod_{i\in\mathcal{P}^r_{n'}} \mathbb{1}(-W_i\mathbf{x}-b_i+U_i \geq 2U_i) = 1\},
$$

134   we have

$$
\begin{aligned}
&f_{PAM}(\mathbf{x})|_{\mathcal{E}} \\
&= \sum_{n'\in\mathcal{N}_D} \prod_{i\in\mathcal{P}^l_{n'}} \mathbb{1}(W_i\mathbf{x}+b_i \geq 0) \prod_{i\in\mathcal{P}^r_{n'}} \mathbb{1}(-W_i\mathbf{x}-b_i \geq 0)\big[\prod_{i\in\mathcal{P}^l_{n'}\cup\mathcal{P}^r_{n'}} 2U_i\big]V(\mathbf{x};\theta_{n'}).
\end{aligned}
$$

135 If we set $W_i$, $b_i$ following the pipeline in Appendix B, we have $\mathcal{N}_D = \mathcal{S}_\Delta$ (see definition in Eq. 3).
136 Then for each $\Delta \in \mathcal{S}_\Delta$, by setting

$$\Big[ \prod_{i \in \mathcal{P}^l_{n'} \cup \mathcal{P}^r_{n'}} 2U_i \Big] V(\mathbf{x}; \theta_{n'}) = W_\Delta \mathbf{x} + b_\Delta,$$

137 we have

$$f_{PAM}(\mathbf{x})|_{\mathcal{E}} = g^L(\mathbf{x}).$$

138 To bound the probability of $Pr(f_{PAM}(\mathbf{x}) = f_{\text{DNN}}(\mathbf{x}))$, we need to bound

$$\begin{aligned}
&Pr(f_{PAM}(\mathbf{x}) = f_{DNN}(\mathbf{x})) \\
&= Pr(\exists n' \in \mathcal{N}_D, \prod_{i \in \mathcal{P}^l_{n'}} \mathbb{1}(W_i\mathbf{x} + b_i + U_i \geq 2U_i) \prod_{i \in \mathcal{P}^r_{n'}} \mathbb{1}(-W_i\mathbf{x} - b_i + U_i \geq 2U_i) = 1) \\
&= \sum_{n' \in \mathcal{N}_D} Pr(\prod_{i \in \mathcal{P}^l_{n'}} \mathbb{1}(W_i\mathbf{x} + b_i \geq U_i) \prod_{i \in \mathcal{P}^r_{n'}} \mathbb{1}(-W_i\mathbf{x} - b_i \geq U_i) = 1).
\end{aligned}$$

(15)

139 According to Eq. 15, $Pr(f_{PAM}(\mathbf{x}) = f_{DNN}(\mathbf{x}))$ increases as $U_i$ decreases. When $U_i = 0$, it's
140 easy to get $Pr(f_{PAM}(\mathbf{x}) = f_{DNN}(\mathbf{x})) = 1$. Therefore, with $U_i \to 0$, we have $Pr(f_{PAM}(\mathbf{x}) =$
141 $f_{DNN}(\mathbf{x})) \to 1$.

142 $\qquad\qquad\qquad\qquad\qquad\qquad\qquad\qquad\qquad\qquad\qquad\qquad\qquad\qquad\qquad\qquad\qquad\qquad\qquad\qquad\qquad\qquad$ $\square$

## F   Proof of Theorem 4

144 Before providing the proof of Theorem 4, we establish Lemma 1 as its foundation.

145 **Lemma 1** *Under Assumption 1 in the main text, for any $p$, $n > 0$, $\epsilon \in (0, 1)$, and $\mathbf{z} \in [0, 1]^{p+n-1}$,*
146 *if we have a function $Q(\mathbf{z}) = z_1 z_2 ... z_{p+n-1}$, a function could be built on the basis of $Q(\mathbf{z})$ which*
147 *can 1) approximates any function from $F_{n,p}$ with an error bound $\epsilon$ in the sense of $L^\infty$ with at*
148 *most $N_Q p^n (N + 1)^p$ parameters, where $N_Q$ is the number of trainable parameter in $Q$, and*
149 $N = \lceil (\frac{n!}{2^p p^n} \frac{\epsilon}{2})^{-\frac{1}{n}} \rceil$.

150 **Proof:**   *By replacing Ep. 18's nested $Q$ with $Q(\mathbf{z})$ in Theorem 1 in [1], we could get the conclusion.*
151 $\square$

152 **Theorem 4** *For any $p$, $n > 0$ and $\epsilon \in (0, 1)$, we have a PAM which can 1) approximates any function*
153 *from $F_{n,p}$ with an error bound $\epsilon$ in the sense of $L^\infty$ with at most $2p^n(N+1)^p(p+n-1)$ parameters,*
154 *where $N = \lceil (\frac{n!}{2^p p^n} \frac{\epsilon}{2})^{-\frac{1}{n}} \rceil$.*

155 **Proof:**   To prove Theorem 4, we first show that there exists a PAM $f_{PAM}(\mathbf{z})$ outputting
156 $z_1 z_2 ... z_{N_{p+n-1}}$. In particular, we construct a $(p + n)$-depth oblique tree $T$. For any $T$'s internal node
157 $i$, we set $U_i = U_C$ as an extremely large hyper-parameter. For any depth $d \in \{1, 2, ..., p + n - 1\}$,
158 all nodes with depth $d$ share the same hyperplane with $b_i = 0$ and

$$W_{i,j} = \begin{cases} 1, j = d, \\ 0, j \neq d, \end{cases}$$

159 for $j \in \{1, 2, ..., p + n - 1\}$, which means that each depth has 2 parameters, and the oblique tree has
160 $2(p + n - 1)$ parameters.

161 According to the Assumption 1 in the main text, $\mathbf{x}$ is bounded, which means that with $U_C \geq 1$, for
162 any leaf node $l$, we have

$$\begin{aligned}
a_{\Delta_l}(\mathbf{z}) &= \prod_{n \in \mathcal{P}^l_l} \min\max((z_{d_l} + U_C, 0), 2U_C) \prod_{l \in \mathcal{P}^r_l} \min\max(-z_{d_l} + U_C, 0), 2U_C) \\
&= \prod_{l \in \mathcal{P}^l_l} z_{d_l} + U_C \prod_{l \in \mathcal{P}^r_l} -z_{d_l} + U_C,
\end{aligned}$$

where $d_n$ is the depth of node $n$. Then for any oblique tree's node $n$, $V(\mathbf{z}; \theta_n)$ is fixed following

$$V(\mathbf{z}; \theta_n) = \begin{cases} 0, & n \text{ is the internal node,} \\ \frac{(-1)^{|\mathcal{P}_n^r|}}{2^{p+n-1}}, & n \text{ is the leaf node,} \end{cases}$$

we have

$$f_{PAM}(\mathbf{z}) = z_1 z_2 ... z_{N_{d+n-1}}.$$

With $Q(\mathbf{z}) = f_{PAM}(\mathbf{z}) = z_1 z_2 ... z_{N_{d+n-1}}$, we get the conclusion following the Lemma 1.

$\square$

## G  Implementation Detail

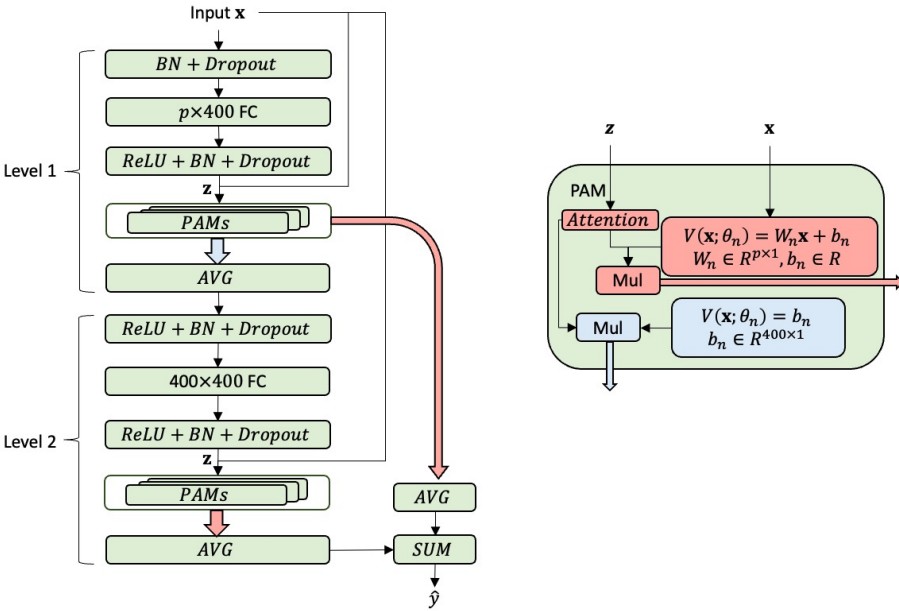

Figure 1: The structure of PAM-Net with 2 levels. BN: Batch Norm Layer; AVG: average; SUM: summation; Dropout: Dropout layer; Mul: matrix multiplication.

As shown in Fig. 1, we combine PAMs in a successive manner similar to cascade forest [2, 3, 4], and name the resultant network as the PAM-Net. In the PAM-Net, each PAM at a higher level calculates interactions among the outputs from its preceding level. Each level of PAM-Net maintains a forest, i.e., a set of PAMs of the same depth $D$, and outputs the average of these PAM outputs.

We follow the principle of Yan et al's work [5] to discuss the complexity of PAMs shown in Fig. 1. In Fig. 1, we consider two kinds of value functions, i.e., $V(x; \theta_n) = W_n x + b_n, W_n \in \mathbb{R}^{p \times 1}, b_n \in \mathbb{R}$. Since a PAM with a D-depth oblique tree has $2^{D-1}$ value functions (1 global value function following Eq. 3 in the main text and $2^{D-1} - 1$ value functions for polyhedrons following Remark 1 in the main text), and the dimension of PAM's input is $p$ according to Eq. 4 in the main text, the memory complexity of these two kinds of value functions is $\mathcal{O}(2^D p)$. In addition to the value function, a D-depth oblique tree has $2^{D-1} - 1$ hyperplanes with $\mathcal{O}(2^{D-1} p)$ trainable parameters. Therefore, the total memory complexity of PAM is $\mathcal{O}(2^D p)$. As for the time complexity, PAM need to 1) calculate the attention score following Eq. 9 in the main text, 2) generate the corresponding values via value functions mentioned above, and 3) output $f_{PAM}$ by multiplying the attention with values following Eq. 8 in the main text. As shown in Table 1, the TIME complexity of PAM is $\mathcal{O}(2^{D-1}(2p + D))$.

For the classification task (Criteo and Avazu dataset), we compress the high dimensional inputs into numerical vectors of a fixed length following the protocol of BARS [6]. For each one-hot encoded or continuous feature, denoted by $x_i^{raw}$, a numerical vector with a fixed length of 10 can be obtained by $\mathbf{W}_i x_i^{raw}$ where $\mathbf{W}_i \in \mathbb{R}^{10 \times |x_i^{raw}|}$ is a trainable embedding matrix. Therefore, for Criteo and Avazu

Table 1: The computation complexity of PAMs in Fig. 1.

| Value Function | Step 1 | Step 2 | Step 3 |
|---|---|---|---|
| $W_n x + b_n$ | $\mathcal{O}(2^{D-1}(p+D))$ | $\mathcal{O}(2^{D-1}p)$ | $\mathcal{O}(2^{D-1})$ |
| $b_n$ | $\mathcal{O}(2^{D-1}(p+D))$ | - | $\mathcal{O}(2^{D-1}p)$ |

datasets, the input of PAMs in the first level of PAM-Net has 390 and 210 elements, respectively. While for the regression task (UK Biobank dataset), we directly use the raw data as the input of the first level, which contains 139 elements.

In PAM-Net, we set the number of levels to 2. A grid search is performed over different configurations of tree depth, i.e. $D = \{4, 5, 6, 7, 8\}$, where the numbers of PAM trees in each level are set to 96, 48, 24, 12, and 6 for the Criteo and Avazu datasets, and 24, 12, 6, 3, and 1 for the UK Biobank dataset, respectively. We conduct grid searches on the dropout rate over $\{0, 0.1, 0.2\}$ and the initial value of $U_i$ over $\{1, 1.5, 2, 2.5, 3\}$. Note that BN and dropout layers were also used in all baseline algorithms and the dropout rate was well-tuned. The Adam optimizer is employed to minimize the loss function using a learning rate of 0.001 with a mini-batch size of 4,096 (Criteo and Avazu datasets) or 1,024 (UK Biobank dataset). To avoid overfitting, we perform early-stopping according to the AUC calculated on the validation set. All algorithms are implemented in PyTorch and tested on servers equipped with Intel Xeon Gold 6150 2.7GHz CPU, 192GB RAM, and an NVIDIA Tesla V100 GPU.