# OpenReview forum: "Polyhedron Attention Module: Learning Adaptive-order Interactions"
_NeurIPS.cc/2023/Conference — NeurIPS 2023 poster_

### Official Review · Reviewer_fykg · 2023-07-02

**Soundness:** 4 excellent
**Presentation:** 3 good
**Contribution:** 3 good
**Rating:** 8
**Confidence:** 3

**Summary:**

This paper presents a feature interaction learning module called the polyhedral attention module (PAM).

The authors show that for any fully ReLU activated DNN, any input x is transformed with respect to a polyhedron defined as the intersection of the halfspace of each layer. Because the input is divided into distinct polyhedrons, the output of a DNN can be written in terms of these polyhedron, which the authors interpret as an attention mechanism.

The authors use this insight to define PAM, which incorporates interaction effects by multiplying the distance from x to each polyhedron. Therefore giving a DNN using PAM an interpretation as a piecewise polynomial function. That is when m activated segments are multiplied, it forms an m degree monomial representing a k-way interaction between the activated segments.

The authors then provide a concept framework for interpreting the learned interaction affects from the PAM. The authors validate the PAM theoretically and empirically on the Criteo, Avazu and the UB Biobank datasets, with PAM achieving the best overall results compared to other models in binary classification for Criteo and Avazu, and brain age prediction on UK Biobank, beating the next best model’s AUC score by 2.2%.


**Strengths:**

* The authors provide an intuitive generalization of the geometric interpretation of DNN into piecewise polynomial function. The theoretical analysis is sound and validated with empirical results.
* Paper is well organized and self contained, with the theoretical concepts being clearly discussed with intuitive supporting figures
The authors provides a general and expressive feature interaction learning model that achieves commendable result across all datasets in classification and regression tasks.

**Weaknesses:**

* Reviewer notices no apparent weaknesses

**Questions:**

* Reviewer has no suggestions for authors

**Limitations:**

Author adequately address limitations of paper.

---

> ### Author Rebuttal · Authors · 2023-08-10
>
> We greatly appreciate your positive review of our work and encouragement and we are open to any additional insights you may have.

---

> > ### Comment · Reviewer_fykg · 2023-08-13
> >
> > Thank you for your response. After reading the other reviews and the authors' responses, I remain confident in my assessment.

---

### Official Review · Reviewer_wPpx · 2023-07-07

**Soundness:** 2 fair
**Presentation:** 2 fair
**Contribution:** 2 fair
**Rating:** 6
**Confidence:** 4

**Summary:**

The paper proposes a Polyhedron Attention Module (PAM) to create piecewise polynomial models to learn feature interactions in multivariate predictive modeling. Specifically, the input space is split into polyhedrons which define the different pieces and on each piece the hyperplanes that define the polyhedron boundary multiply to form the interactive terms, resulting in interactions of adaptive order to each piece. Theoretical analysis and experimental verification are provided to demonstrate the superior of PAM.

**Strengths:**

1. The paper is written carefully with mathematical details. The tables and figures are quite clear.
2. The paper provides both theoretical and experimental verifications for the proposed method.


**Weaknesses:**

1. The paper is quite difficult to follow. I would suggest the authors use more natural language for the analysis of the proposed method.
2. A figure for an overview description of the proposed method is lacking.

**Questions:**

1. The intuition of PAM is that the authors believe data instances belonging to different pieces may endorse interactions in different ways. Why should this be the case? It would be more convincing if the authors could provide real-world examples to illustrate why we should consider data interactions at a local scale.

2. The common drawback of local methods is that inference can be expensive. For PAM's case, the heavy cost may lie in the step of generating polyhedrons for each data instance. How does tree search help reduce this cost?

3. Why the polyhedrons should overlap? An ablation study between overlapping and non-overlapping polyhedrons will be beneficial.

4. (Minor) The paragraph right after Eq. (5) is unfinished.

5. The improvements on Criteo and Avazu datasets are incremental with around 0.1% AUC increase. Although the authors have compared with the improvements of the second-best method, it would be more convincing if the authors could conduct statistical significance tests for PAM's improvements.

6. Overall, the standard deviations of PAM in the experiments are lower than other methods. Can the authors provide a thorough explanation for this improvement?

7. An ablation study on training & inference time of PAM compared to other methods may be beneficial.

I would happily increase my score if the authors carefully tackled my concerns above.


**Limitations:**

The authors have not included a limitation section.

---

> ### Author Rebuttal · Authors · 2023-08-10
>
> Thank you for your follow-up questions and comments. Below we address each question and minor questions in a point by point fashion.
>
> Weaknesses:
>
> Response: The mathematical derivation can develop the approach rigorously and helps demonstrate the solid foundation of PAM, but we agree that it makes our paper uneasy to comprehend. We will rewrite Sections 2 and 3 (with a notation table in the appendix to link the different symbols) and importantly, we will use more plain explanation and revise the current Figure 1 for better demonstration and make it Figure 2. Then, as suggested by reviewer, we will add in an overall figure (new Figure 1) to demonstrate the main idea of PAM. (Please see Fig. S1 in the attached pdf and we welcome more suggestions on revising this overall figure).
>
> Questions:
>
> Response to 1: We utilize the real-world example in [1] to illustrate why different input regions (pieces or polyhedrons) may endorse different interactions.
>
> [1] Li, Zeyu, et al. "Interpretable click-through rate prediction through hierarchical attention." Proceedings of the 13th International Conference on Web Search and Data Mining. 2020.
>
> A record movie genre = horror, user age = young, time = morning has conflicting factors: the combination of the first two encourages the user to watch the movie, whereas the combination of the latter two discourages it because movie-watching usually happens at night. Therefore, rather than capturing the global interaction among these three features, in which user age oppositely impacts on people's movie-watching behavior, categorizing individuals into two local groups based on their preference for movie-watching time or genre, and fitting the interaction weights of "movie genre $\times$ user age" and "time$\times$ user age" separately will achieve better performance.
>
> Response to 2: We employ the oblique tree to search for appropriate polyhedrons in PAM because it could generate polyhedrons which cannot be obtained via simply using independent hyperplanes. As the example shown in Figs. S3a-S3d in the attached pdf, the oblique tree shown in Fig. S3a could divide the input space into three polyhedrons (P1, P2 and P3 in Fig. S3b). As shown in Fig. S3c, this division can not be obtained by two independent hyperplanes. If we split the input space using the hyperplanes H1 and H2, we will get four polyhedrons (see P1, P2, P3 and P4), which is different from those in Fig. S3b. On the contrary, as proved in our paper's Base Case of Appendix B, the oblique tree can identify all possible polyhedrons created by separating hyperplanes but not vice versa. For example, polyhedrons in Fig. S3c can be generated by the oblique tree demonstrated in Fig. S3d.
>
> With $L-1$ hyperplanes (more precisely, truncated hyperplanes with $M$ parameters for each), an oblique tree can generate $2L-2$ polyhedrons. With $L$ value functions (with M parameters for each, $L-1$ for the oblique tree and $1$ for the global value function, see Eq. 4 and Remark 1 in the main text), an oblique tree results in $(2L-1)M$ trainable parameters. If we use the same hyperplanes found by the oblique tree to split the input space, it can create $2^{L-1}$ polyhedrons. It requires totally $(2^{L-1}+L)M$ trainable parameters to train the model. In this sense, the oblique tree does reduce the trainable parameters and computation cost.
>
> Response to 3: We have already conducted the ablation study comparing overlapping and non-overlapping polyhedrons in our paper (refer to Fig. 4c bar for "w/o OP"). We observed that overlapping polyhedrons did improve performance. This might be partially due to two reasons. Firstly, a sample can belong to multiple clusters simultaneously in real life, like a red and green apple may be in red and green groups. Treating a polyhedron as a cluster of input instances, the overlapping ones lead to a fuzzy clustering framework. There is also a practical issue when a sample can only belong to one polyhedron, which results in one and only one non-zero attention score (as per Eq. 3). Model parameters are trained based on gradients of the loss function, and a zero $a_{\Delta}$ causes zero gradients for the value function $V_{\Delta}$ in PAM (as per Eq. 4 and the chain rule), each training iteration will update solely the value functions presented in a leaf and its ancestor nodes in the oblique tree. Secondly, identifying the exact boundary between two polyhedrons is challenging and prone to errors. With overlapping polyhedrons, the actual boundary may lie in that overlapping buffer zone to make the method more robust.
>
> Response to 4: Thank you for pointing out the unfinished sentence after Eq. 5. That half sentence should be removed.
>
> Response to 5: It is a great suggestion to conduct statistical significance tests. We performed more runs (5 additional) of PAM and baseline methods so we have enough data to conduct unpaired t-tests. This test result indicates that PAM significantly outperforms baseline algorithms on all Criteo, Avazu and UK Biobank datasets ($P<0.05$, respectively, as shown in Table S2 in the attached pdf). We will include this result to Appendix.
>
> Response to 6: We did observe this interesting result. It did perform more stably than many of the baselines. It may be because the same PAM module (the same architecture) is stacked (see Figure 1 in Appendix G), which could reduce the variance of the model’s performance according to existing research [2].
>
> [2] Fahmy, Hesham Ahmed, et al. "An ensemble multi-stream classifier for infant needs detection." Heliyon 9.4 (2023).
>
> Response to 7:  Following the reviewer's suggestion, we now added a new comparison of training and inference time. The new results are included in  Table S1 in the attached PDF file to compare the inference time of PAM with those of baseline methods. The training time comparison can be found in Table 2 in the original paper.

---

> > ### Comment · Reviewer_wPpx · 2023-08-13
> > **Score Update**
> >
> > I am satisfied with the authors' careful responses to my concerns and have increased my score accordingly. Good luck!

---

### Official Review · Reviewer_8xSi · 2023-07-07

**Soundness:** 3 good
**Presentation:** 3 good
**Contribution:** 3 good
**Rating:** 6
**Confidence:** 3

**Summary:**

The ReLU-activated DNN will split the input space into pieces such as polyhedrons. The author proposes a polyhedron attention module (PAM) to capture the interaction between different pieces of input spaces. And they propose an approximation theorem for PAM. The polyhedrons are generated via an oblique tree, and each tree node means a sub-space of a polyhedron. And for each node, two functions will be learned: the splitting function which decides how to further split the hyperplane, and the value function. The authors also show that such a module needs fewer parameters than the plain DNN.

**Strengths:**

1: The author provides a detailed and clear explanation on how the proposed works.

2: The general idea is novel. The split and attention step can dynamically capture the adaptive interaction between different data instances.

3: The author also provides necessary justification for the methods.

**Weaknesses:**

1: Since I am not familiar with the baseline proposed in the paper, in section 6.3, I want to see whether other methods can find the same effects and interactions or not.

2: Table 2 still shows that the PAM has more parameters and more running time compared to other methods. It will be helpful if the author can add a few explanations.

**Questions:**

N/A

---

> ### Author Rebuttal · Authors · 2023-08-10
>
> Thank you for your follow-up questions and comments. Below we address each question and minor questions in a point by point fashion.
>
> Weaknesses:
>
> Response to 1: Reviewer asked about the baseline methods in Section 6.3 and whether these methods can find the same effects and interactions. As far as we know, our interpretation framework introduced in Section 4 is the first algorithm to extract the main and interaction effects from DNN models. We do stress that although it was developed for PAM, it can be applied to those DNN architectures where activation functions are piece-wise linear (e.g., ReLU, ReLU6, and HardTanh, please also consult with the proof in Appendix D).
>
> After carefully checking all baseline methods in Section 6, we found that our interpretation framework could be used to extract interactions from AOANet and DCN. The main effects of precentral gyrus and thalamus identified by PAM were found by DCN and AOANet, respectively (Fig. S2 in the attached pdf), and all three algorithms found the two-way interactions between the left and right frontal pole. In addition to the shared top-5 brain regions identified by these three algorithms, PAM additionally identified the main effect of the lateral occipital cortex and frontal pole and two-way interaction effects related to the insular and subcallosal cortex regions.
> We will put these results into section 6.3.
>
> Response to 2: It does make sense to explain why PAM has more parameters and run time in the experiments. Note that our theoretical analysis shows that PAM has more parameter efficiency for universal approximation than ReLU-activated DNN. In other words, ReLU-activated DNN may need more parameters to reach universal approximation. However, it does not prevent the overfitting of the larger size DNN on a specific task. In our experiments, the standard DNN model (ReLU-activated), if using more parameters following Theorems 3 and 4, produced worse test performance than the one we reported.  For fair comparison, we reported on the architectures of all the other models which gave the best performance. It shows that these models used slightly smaller numbers of parameters, but their best tuned performance was worse than that of PAM (Fig. 4a).
>
> For experiments on Criteo and Avazu, although PAM has slightly more trainable parameters, its run time is comparable with that of the baselines with fewer trainable parameters (i.e., DESTINE and DCN-V2) because all value functions within the red box in Fig.1 in Appendix G have the same input $x$ and can thus be calculated in parallel.
>
> We would also like to point out that we used pre-existing modules within the PyTorch package to implement PAM. These modules have well-optimized CUDA kernels for forward and backward functions, leading to efficient execution of DNN. However, the oblique tree data structure is not included in PyTorch. We believe that a customized CUDA kernel would further enhance the run time of PAM.
>
> We will add these explanations to a supplemental section.

---

> > ### Comment · Reviewer_8xSi · 2023-08-18
> > **Rebuttal read**
> >
> > Thanks for your response and clear explanation. They addressed my questions. I would like to keep my score.

---

### Official Review · Reviewer_61Ki · 2023-07-11

**Soundness:** 3 good
**Presentation:** 2 fair
**Contribution:** 2 fair
**Rating:** 6
**Confidence:** 3

**Summary:**

The paper introduces a more general nonlinearity, PAM, to better capture the data features' interactions. Theoretical justification and interoperability are performed along with empirical results.

**Strengths:**

The method seems to be inspired from sharp mathematical observations. It has mathematical interpretation and a sound rational connecting existing methods.

**Weaknesses:**

The explanation and intuition behind the methods are unclear; for example, the introduction of k-way interaction is put in front of the article but lacks a description of what it means and why it is essential.
It is unclear how the method generalizes to complicated models and what the complexity will be.

**Questions:**

It will be nice to simplify the notations and convey the concepts more plainly. The article can improve by reorganization. More experiments can also help to validate the method.

**Limitations:**

Limitations have been properly addressed.

---

> ### Author Rebuttal · Authors · 2023-08-10
>
> Thank you for your follow-up questions and comments. Below we address each question and minor questions in a point by point fashion.
>
> Weakness:
>
> Response: We further explain the rational of the presentation of our paper here and will revise our paper to explain better with examples. Feature interactions can occur in many different forms. The k-way interaction is defined first (according to existing work) to provide us a mathematical basis so we can derive the quantitative algorithm in the subsequent sections. K-way interactions mean any kind of interaction among k different input features. A deep learning model with highly nonlinear activation function, for a simple example, $sigmoid(\sum w_i x_i)$ with $i = 1, \cdots, d$ specifies a d-way interaction of any arbitrary order (as $sigmoid$ has derivatives of arbitrary order), and lack of interpretability. Instead, a lower-order interaction among fewer features (lower way) can significantly improve the model explainability.  For a simple example, by incorporating the two-way interaction effect between gender (0/1: female and male) and age into the linear regression model  $height\sim w_1\cdot gender+w_2\cdot age+w_3\cdot gender\times age+w_0$ (where $w_0,w_1,w_2$ and $w_3$  are trainable parameters), the effects of female's age on the heights will be different from that of male ($w_2$ v.s. $w_2+w_3$). Our approach seeks to identify such low-way and low-order interactions for better interpretability.
>
> The most straightforward way to define k-way interactions is to form multiplicative terms among the k features (e.g., $x_1, x_2, \cdots x_k$). Our attention module PAM adaptively derives multiplications of linear functions (corresponding to the boundary hyperplanes of a polyhedron) when learning the partition of the input space into polyhedrons. Thus, the linear function multiplication generates multiplicative terms among various features (being adaptive to different input regions/polyhedrons) in generally lower order polynomials.
>
> Response: The following answers the second critique about how to generalize PAM to complicated models and the complexity of PAM.
> As mentioned in Section 6 and Appendix G, the proposed PAM can be used as a basic module in a stack to generalize to complicated models. Each module maps its own input (output from early PAM blocks) according to Eq.4. PAM has already been used as the basic module in our experiments to build a complicated model whose architecture is given in Appendix G.
>
> The computation complexity of other deep learning methods has been previously discussed. We follow the principle of Yan et al. 2022 [1] to now discuss the complexity of PAM. As shown in Eq. 4, the dimension of PAM's input is $p$. In our experiment, we consider two kinds of value functions in this work, i.e., $V(x;\theta_n)=W_nx+b_n, W_n\in\mathbb{R}^{p\times 1}, b_n\in\mathbb{R}$ and $V(x;\theta_n)=b_n$, $b_n\in\mathbb{R}^p$. Since a PAM with a D-depth oblique tree has $2^{D-1}$ value functions (1 global value function following Eq. 3 and $2^{D-1}-1$ value functions for polyhedrons following Remark 1), the memory complexity of these two kinds of value functions is $\mathcal{O}(2^{D-1}p)$. In addition to the value function, a D-depth oblique tree has $2^{D-1}-1$ hyperplanes with $\mathcal{O}(2^{D-1}p)$ trainable parameters. Therefore, the total MEMORY complexity of PAM is $\mathcal{O}(2^{D}p)$.
> As for the time complexity, PAM need to 1) calculate the attention score following Eq. 9, 2) generate the corresponding values via value functions mentioned above, and 3) output $f_{PAM}$ by multiplying the attention with values following Eq 8. The computation complexity is shown as follows:
>
> | Value function | Step 1  |Step 2  |Step 3 |
> | ---- | ----------- |----------- |----------- |
> | $W_nx+b_n$ |$ \mathcal{O}(2^{D-1}(p+D)) $      |$\mathcal{O}(2^{D-1}p)$       |$\mathcal{O}(2^{D-1})$       |
> |  $b_n$   | $\mathcal{O}(2^{D-1}(p+D))$        |-       |$\mathcal{O}(2^{D-1}p)$        |
>
> Thus, the TIME complexity of PAM is $\mathcal{O}(2^{D-1}(2p+D))$. Note that in our experiment $p\gg D$. We will add this analysis to Remark 1 to illustrate the complexity of our model.
>
> [1] Yan, Bencheng, et al. "APG: Adaptive parameter generation network for click-through rate prediction." Advances in Neural Information Processing Systems 35 (2022): 24740-24752.
>
> Questions:
>
> Response: We will simplify the notations and provide a notation table to link the different symbols. We will demonstrate the concepts with an overall figure as suggested by Reviewer wPpx (please see Fig. S1 in the attached pdf and response to Reviewer wPpx).
>
> More experiments have been performed and added to validate our method (also in response to other reviewers' suggestions):
>
> a) As shown in Table S1 in the attached pdf, the inference time of PAM and other baseline methods is compared.
>
> b) As shown in Table S2 in the attached pdf, the unpaired t-test has been conducted to evaluate the statistical significance of PAM's improvement over baselines (using the performance numbers in Figure 4a and additional runs of the different models).

---

### Author Rebuttal · Authors · 2023-08-10

We appreciate all four reviewers for their positive and encouraging summary about our paper's strength. All have noted that our paper provides a sound, novel approach with theoretical justification and empirical verification (e.g., from reviewer 61Ki, "sharp mathematical observations", "mathematical interpretation and a sound rational connecting existing methods"; from reviewer 8xSi, "the idea is novel"; from reviewer wPpx, "provides both theoretical and experimental verification", from reviewer fykg, "thereotical concepts being clearly discussed with intuitive supporting figures", "achieve commendable results"). Three reviewers suggested more experiments to either clarify baseline methods or further validate the proposed approach. We have performed all advised experiments and explain the results in the following point-by-point responses to all weaknesses arisen from the critiques which help us improve the quality of our paper.

---

### Decision · Program_Chairs · 2023-09-21

**Decision:**

Accept (poster)

**Comment:**

In this paper, the authors first explain the intuition of Deep ReLU networks as cutting the input space into polyhedrons and learning affine functions within each polyhedron. Then the authors use this intuition to build the proposed PAM network module, by cutting the input space into polyhedrons with oblique tree, and then parameterising the value functions associated with the polyhedrons as well as their interactions. Both theoretical analysis on expressiveness as well as empirical evaluations on click-through-rate prediction & medical dataset are provided.

Reviewers agree that the proposed approach is novel and the justifications (theoretical and empirical) are overall sound. In author feedback the authors provided further clarifications and experiments that also help enhance the paper. I would suggest the authors to include them in the final manuscript.

After a brief reading of the paper, I feel that while I agree with all the reviewers' recommendation of acceptance, I was wondering whether the use of k-way interaction motivation may also lead to some interesting further discussions. The PAM method, while is claimed to capture all the k-way interactions, does not do so explicitly compared to e.g., constructing polynomials. In this regard, I recommend the authors to discuss the line of work on (Generalised) Neural Additive Models and clarify the interpretability properties of PAM.